# Deep Double Descent via Smooth Interpolation

**Matteo Gamba**                                                        *mgamba@kth.se*
*KTH Royal Institute of Technology*

**Erik Englesson**                                                      *engless@kth.se*
*KTH Royal Institute of Technology*

**Mårten Björkman**                                                     *celle@kth.se*
*KTH Royal Institute of Technology*

**Hossein Azizpour**                                                    *azizpour@kth.se*
*KTH Royal Institute of Technology*

**Reviewed on OpenReview:** *https://openreview.net/forum?id=fempQstMbV*

## Abstract

The ability of overparameterized deep networks to interpolate noisy data, while at the same time showing good generalization performance, has been recently characterized in terms of the double descent curve for the test error. Common intuition from polynomial regression suggests that overparameterized networks are able to sharply interpolate noisy data, without considerably deviating from the ground-truth signal, thus preserving generalization ability.

At present, a precise characterization of the relationship between interpolation and generalization for deep networks is missing. In this work, we quantify sharpness of fit of the training data interpolated by neural network functions, by studying the loss landscape w.r.t. to the input variable locally to each training point, over volumes around cleanly- and noisily-labelled training samples, as we systematically increase the number of model parameters and training epochs. Our findings show that loss sharpness in the input space follows both model- and epoch-wise double descent, with worse peaks observed around noisy labels. While small interpolating models sharply fit both clean and noisy data, large interpolating models express a smooth loss landscape, where noisy targets are predicted over large volumes around training data points, in contrast to existing intuition [1].

## 1 Introduction

The ability of overparameterized deep networks to interpolate noisy data, while at the same time showing good generalization performance (Belkin et al., 2018; Zhang et al., 2018), has been recently characterized in terms of the double descent curve of the test error (Belkin et al., 2019; Geiger et al., 2019). Within this framework, as model size increases, the test error follows the classical bias-variance trade-off curve (Geman et al., 1992), peaking as models become large enough to perfectly interpolate the training data, and decreasing as model size grows further (Belkin et al., 2019). This phenomenon, largely studied in the context of regression (Bartlett et al., 2020; Muthukumar et al., 2020) and random features (Belkin et al., 2020), at present lacks a precise characterization relating interpolation to generalization for deep networks.

Current intuition from linear and polynomial regression suggests that, under some hypothesis on the training sample, large overparameterized models are able to perfectly interpolate both cleanly- and noisily-labeled samples, without considerably deviating from the ground-truth signal, thus showing good performance despite overfitting the training data (Muthukumar et al., 2020; Bartlett et al., 2020; Nakkiran et al., 2019a).

---

[1]Source code to reproduce our results available at `https://github.com/magamba/double_descent`

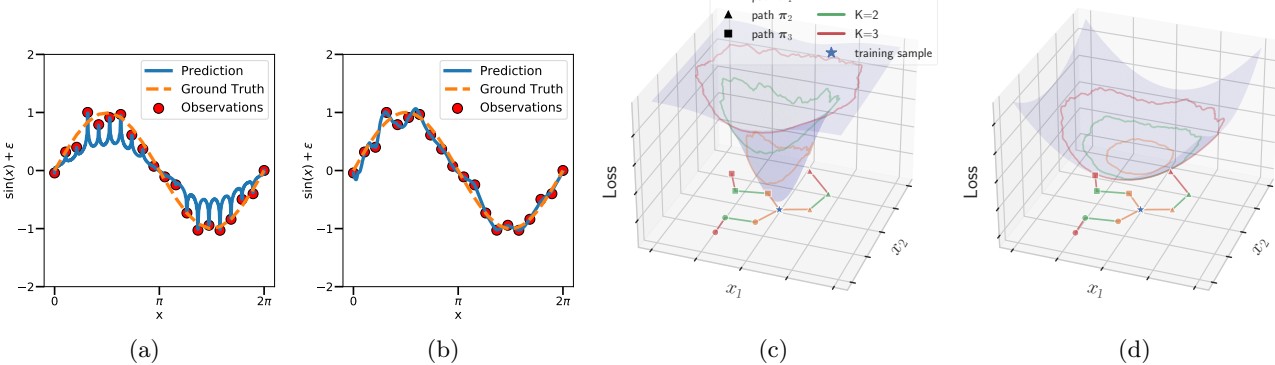

Figure 1: **Intuition from overparameterized regression.** a) Polynomial of large degree, trained with gradient descent to fit noisy scalar data, reproducing the polynomial regression experiment of Nakkiran et al. (2019a), and reflecting common intuition on double descent, suggesting that the generalization ability of large interpolating models is tied to sharply fitting of noisy data, thus resulting in models that do not deviate considerably from the ground truth signal. b) In this work we show that, contrary to intuition, deep networks *smoothly* interpolate both clean and noisy data, and that improved generalization in the interpolating regime is tied to smoothness of the loss w.r.t. the input variable. **Geodesic MC integration.** For each base training point, we generate $P$ geodesic paths by connecting a sequence of augmentations of increasing strength, which we use to cover volumes of increasing size in the loss landscape around each training point. We compare points that are c) sharply interpolated from those that are d) smoothly interpolated.

Figure 1a illustrates this phenomenon, showing a polynomial of large degree that perfectly fits the training data, with predictive function sharply interpolating noisy samples (intuitively corresponding to a spike at each training point), while overall remaining close to the data-generating function.

In this work, we study the emergence of double descent for the test error of deep networks (Nakkiran et al., 2019b) through the lens of smoothness of interpolation of the training data, as model size as well as the number of training epochs vary, for models trained in practice. To quantify smoothness of interpolation, we conduct an empirical exploration of the loss landscape w.r.t. the input variable, by providing explicit measures of sharpness of the loss, focusing on image classification.

Due to the inherently noisy nature of Euclidean estimators in pixel space, and following the *manifold hypothesis* Pope et al. (2020); Bengio (2013); Narayanan & Mitter (2010), postulating that natural data lies on a combination of manifolds of lower dimension than the input data's ambient dimension, we constrain our measures to the support of the data distribution, locally to each training point.

Our empirical study shows that the polynomial intuition in Figure 1a does not hold in practice for deep networks, which instead smoothly interpolate both clean and noisy data (Figure 1b). Specifically, smooth interpolation – emerging both for large overparameterized networks and prolonged training – results in large models confidently predicting the (noisy) training targets over large volumes around each training point.

**Contributions**

- We present the first systematic empirical study of smoothness of the loss landscape of deep networks in relation to overparameterization and interpolation for natural image datasets.

- Starting from infinitesimal smoothness measures from prior work, we introduce volumetric measures that capture loss smoothness when moving away from training points.

- We develop a geodesic Monte Carlo integration method for constraining our measures to a local approximation of the data manifold, in proximity of each training point.

- We present an empirical study of model-wise and epoch-wise double descent for neural networks trained without confounders (explicit regularization, data augmentation, batch normalization), as well as for commonly-found training settings. By decoupling smoothness from generalization, we empirically show that overparameterization promotes input-space smoothness of the loss landscape. Particularly, we produce practical examples in which smoothness of the learned function of deep networks does not result in improved generalization, highlighting that the implicit regularization effect of overparameterization should be studied in terms of reduced variation of the learned function.

## 2 Related work

Recent years have seen increased interest in the study of smoothness of deep networks in relationship to generalization. For studies of *learned representations*, interpreting networks as functions of their parameters, loss landscape smoothness has been related to improved generalization (Ma & Ying, 2021; Foret et al., 2020; Rosca et al., 2020), increased stability to perturbations (Keskar et al., 2017), reduced minimum description length (Hochreiter & Schmidhuber, 1997), as well as better model compression (Chang et al., 2021).

Additionally, for networks interpreted as functions of their input, for a *fixed parameterization*, sensitivity of the networks' learned function has been connected to generalization performance (LeJeune et al., 2019; Novak et al., 2018). Indeed mounting evidence, both empirical (Gamba et al., 2022; 2020; Novak et al., 2018) as well as theoretical (Bubeck & Sellke, 2021; Neyshabur et al., 2018), suggests that large overparameterized models achieve robust generalization (Ma & Ying, 2021) via smoothness of the learned function. While overparameterization alone is not enough to guarantee strong robustness (Chen et al., 2021; Rice et al., 2020), the large number of parameters of modern networks is thought to promote implicit regularization of the network's function (Gamba et al., 2022; Bubeck & Sellke, 2021; Neyshabur et al., 2018; 2015). In this context, a direct study of interpolation via the parameter-space interpretation is limited by confounders, such as symmetries of linear layers (Singh & Jaggi, 2020; Li et al., 2015), for which different parameterizations may yield the same equivalent interpolating function (Simsek et al., 2021). Thus, our work adopts the input-space view of the loss landscape, to directly study sharpness of interpolation around each training point.

Our methodology builds upon input-space sensitivity analyses for neural networks, presenting a first systematic study of the role of overparameterization in promoting smoothness of the network's learned function. The smoothness measures presented in section 3, are inspired by the vast body of work on the loss landscape of neural networks in parameter space. Due to the extensive theoretical literature on double descent in simplified controlled settings such as linear regression (Muthukumar et al., 2020; Bartlett et al., 2020; Belkin et al., 2018), in the following we mainly draw connections to prior work targeting deep networks.

**Deep Double Descent**  Double descent (Belkin et al., 2019; Geiger et al., 2019) was first observed for several machine learning algorithms for increasing model size (model-wise). Later, Nakkiran et al. (2019b) showed a similar trend during training of deep networks (epoch-wise), as well as w.r.t. dataset size (sample-wise). The phenomenon has been studied from various perspectives: bias-variance decomposition (Yang et al., 2020; Neal et al., 2018), samples to parameters ratio (Belkin et al., 2020), parameter norms (Belkin et al., 2019), and decision boundaries (Somepalli et al., 2022).

In this work, we study model-wise and epoch-wise double descent in terms of smoothness of the loss landscape with respect to the input, and separate the analysis in terms of clean and noisily-labeled data points. Importantly, in contrast to existing studies of double descent (Belkin et al., 2020), we focus on input space and on the training loss – a quantity that does not follow double descent – and study sharpness metrics based on training data, showing that they strongly correlate with the test error.

The most related work to ours is the concurrent one of Somepalli et al. (2022) studying decision boundaries in terms of reproducibility and double descent. Our works differ in that we study double descent in the loss landscape. Furthermore, our study takes a closer look at the impact of clean and noisily-labeled samples, and presents settings in which the emergence of regularity of the loss landscape does not result in improved generalization. Finally, we focus on implicit regularization (by disabling batch norm, data augmentation) and a simpler optimization procedure (constant learning rate SGD) to reduce confounding factors.

**Loss Landscape of Neural Networks**  To understand the remarkable generalization ability of deep networks (Xie et al., 2020; Geiger et al., 2019; Keskar et al., 2017), as well as to design better training criteria (Foret et al., 2020), several works study the loss landscape of deep networks in *parameter space*, focusing on solutions obtained by SGD (Kuditipudi et al., 2019), as well as the optimization process (Arora et al., 2022; Li et al., 2021). Inspired by such literature, we quantify smoothness of the loss landscape by estimating the *sharpness* of the loss, as proposed by Foret et al. (2020) and Keskar et al. (2017) for the parameter-space, but we perform our analysis in *input-space*. Importantly, in this work we focus on image classification tasks, and study smoothness of interpolation of training data points.

**Input Space Sensitivity and Smoothness**  Novak et al. (2018) present an empirical sensitivity study of fully-connected networks with piece-wise linear activation functions through the input-output Jacobian norm, which is shown to strongly correlate to the generalization ability of the networks considered. Their study proposes an infinitesimal analysis of the Jacobian norm at training and validation points, as well as the use of input-space trajectories in proximity of the data manifold to probe trained networks. LeJeune et al. (2019) analyse second-order information (the tangent Hessian of a neural network) by using weak data augmentation to constrain their measure to the proximity of the data manifold. Lastly, Gamba et al. (2022) introduce a nonlinarity measure for piece-wise linear networks, that strongly correlates with the test error in the second descent for large overparameterized models. Similar to the first two works, we study smoothness of neural networks, using the Jacobian and Hessian norm of neural networks trained in practice, and similar to the latter work, we provide a systematic study of double descent, which we further extend to epoch-wise trends.

Finally, Rosca et al. (2020) postulate a connection between model-wise double descent and smoothness: during the first ascent models fit the training data at the expense of smoothness, while the second descent happens as the model size becomes large enough for smoothness to increase.

Later, Bubeck & Sellke (2021) theoretically prove a universal law of robustness highlighting a trade-off between the model size and the Lipschitz constant of a learning algorithm w.r.t. its input variable. Our work provides empirical evidence supporting the postulate of Rosca et al. (2020) and the law of robustness of Bubeck & Sellke (2021).

## 3 Methodology

Our leading research question is to quantify smoothness of interpolation of training data for deep networks trained on classification tasks, as the number of model parameters is increased. We interpret a network as a function with input variable $\mathbf{x} \in \mathbb{R}^d$ and learnable parameter $\boldsymbol{\theta}$, incorporating all weights and biases. Our study focuses on the landscape of the loss $\mathcal{L}_{\boldsymbol{\theta}}(\mathbf{x}, y) := \mathcal{L}(\boldsymbol{\theta}, \mathbf{x}, y)$ treated as a function of the input $\mathbf{x}$, with target $y$. Inspired by the literature on the loss landscape of neural networks in parameter space (Foret et al., 2020; Dinh et al., 2017; Keskar et al., 2017), we quantify (the lack of) smoothness by devising explicit measures of loss sharpness in a neighbourhood of training points $(\mathbf{x}_n, y_n)$, for $n = 1, \dots, N$. Crucially, for any given network, we focus on sharpness w.r.t. the input variable $\mathbf{x}$, keeping the parameter $\boldsymbol{\theta}$ fixed.

We begin by describing infinitesimal sharpness in section 3.1, which we compute in proximity of the data manifold local to each training point in section 3.2. Finally, we introduce a method for estimating sharpness over data-driven volumes by exploiting data augmentation in section 3.3, and in section 3.4 we detail the chosen data augmentation strategies. The proposed methodology enables us to measure sharpness of interpolation of the training data, by restricting our study near the support of the data distribution.

### 3.1 Sharpness at Data Points

To estimate how sharply the loss changes w.r.t. infinitesimal perturbations of the input variable $\mathbf{x}$, we study the Jacobian of the loss,

$$\mathbf{J}(\mathbf{x}, y) := \frac{\partial}{\partial \mathbf{x}} \mathcal{L}_{\boldsymbol{\theta}}(\mathbf{x}, y) \tag{1}$$

To measure sharpness at a point $(\mathbf{x}_n, y_n)$, we follow Novak et al. (2018), and compute the $\ell_2$ norm of $\mathbf{J}(\mathbf{x}_n, y_n)$, which we take in expectation over the training set $\mathcal{D} = \{(\mathbf{x}_n, y_n)\}_{n=1}^N$,

$$J = \mathbb{E}_{\mathcal{D}} \|\mathbf{J}(\mathbf{x}, y)\|_2 \tag{2}$$

assuming that the loss is differentiable one time at the points considered. Intuitively, sharpness is measured by how fast the loss $\mathcal{L}_{\boldsymbol{\theta}}(\mathbf{x}, y)$ changes in infinitesimal neighbourhoods of the training data, and a network is said to smoothly interpolate a data point $\mathbf{x}_n$ if the loss is approximately flat locally around the point and the point is classified correctly according to the corresponding target $y_n$. Throughout our experiments, the Jacobian $\mathbf{J}$ is computed using a backward pass w.r.t. the input variable $\mathbf{x}$.

Equation 2 provides first-order information about the loss landscape. To gain knowledge about curvature, we also study the Hessian of the loss w.r.t. the input variable,

$$\mathbf{H}(\mathbf{x}, y) := \frac{\partial^2}{\partial \mathbf{x} \partial \mathbf{x}^T} \mathcal{L}_{\boldsymbol{\theta}}(\mathbf{x}, y) \tag{3}$$

whose Frobenius norm again we take in expectation over the training set

$$H = \mathbb{E}_{\mathcal{D}} \|\mathbf{H}(\mathbf{x}, y)\|_2 \tag{4}$$

The Hessian tensor in Equation 3 depends quadratically on the input space dimensionality $d$, providing a noisy Euclidean estimator of loss curvature in proximity of the input data. Following the *manifold hypothesis* (Bengio, 2013; Narayanan & Mitter, 2010), stating that natural data lies on subspaces of dimensionality lower than the ambient dimension $d$, we restrict Hessian computation to the tangent subspace of each training point $\mathbf{x}_n$. Starting from Equation 1, throughout our experiments, Equation 3 is estimated by computing the tangent Hessian, as outlined in the next section.

### 3.2 Tangent Hessian Estimation

To constrain Equation 3 to the support of the data distribution, we adapt the method by LeJeune et al. (2019) and estimate the loss Hessian norm projected onto the data manifold local to each training point.

For any input data point $(\mathbf{x}_n, y_n)$ and corresponding Jacobian $\mathbf{J}(\mathbf{x}_n, y_n)$, we generate $M$ augmented data points $\mathbf{x}_n + \mathbf{u}_m$ by randomly sampling a displacement vector $\mathbf{u}_m$ using weak data augmentation. For each sampled $\mathbf{u}_m$, we then estimate the Hessian $\mathbf{H}(\mathbf{x}_n, y_n)$ projected along the direction $\mathbf{x}_n + \mathbf{u}_m$, by computing the finite difference $\frac{1}{\delta}\mathbf{J}(\mathbf{x}_n, y_n) - \mathbf{J}(\mathbf{x}_n + \delta \mathbf{u}_m, y_n)$. Then, following Donoho & Grimes (2003) we estimate the Hessian norm directly by computing

$$H = \frac{1}{M^2 \delta^2} \mathbb{E}_{\mathcal{D}} \Big( \sum_{m=1}^{M} \|\mathbf{J}(\mathbf{x}_n, y_n) - \mathbf{J}(\mathbf{x}_n + \delta \mathbf{u}_m, y_n)\|_2^2 \Big)^{\frac{1}{2}} \tag{5}$$

which is equivalent to a rescaled version of the rugosity measure of LeJeune et al. (2019). Importantly, different from rugosity, we generate augmentations $\mathbf{x}_n + \mathbf{u}_m$ by using weak colour transformations in place of affine transformations (1-pixel shifts), since weak photometric transformations are guaranteed to be fully on-manifold. Details about the specific colour transformations are presented in appendix C.

### 3.3 Sharpness over Data-Driven Volumes

The measures introduced in Equations 2 and 5, capture local sharpness over infinitesimal neighbourhoods of input data points. To study how different networks fit the training data, we devise a method for estimating loss sharpness over volumes centered at each training point $\mathbf{x}_n$, as one moves away from the point. Essentially, we exploit a variant of Monte Carlo (MC) integration to capture sharpness over data-driven volumes, by applying two steps. First, we integrate the Jacobian and Hessian norms along geodesic paths $\boldsymbol{\pi}_p \subset \mathbb{R}^d$ based at $\mathbf{x}_n$, on the data manifold local to each training point, for $p = 1, \ldots, P$. Second, we estimate sharpness over the volume covered by the loss along the $P$ paths via MC integration. The following details each step.

**Sharpness along geodesic paths** For each training point $(\mathbf{x}_n, y_n) \in \mathcal{D}$, we aim to estimate loss sharpness as we move away from $\mathbf{x}_n$, while traveling on the support of the data distribution. To do so, we exploit a sequence of weak data augmentations of increasing strength to generate $P$ paths $\boldsymbol{\pi}_p \subset \mathbb{R}^d$ in the input space, each formed by connecting augmentations of $\mathbf{x}_n$ in order of increasing strength.

Formally, let $\mathcal{T}_\mathbf{s} : \mathbb{R}^d \to \mathbb{R}^d$, represent a family of smooth transformations (data augmentation) acting on the input space and governed by parameter $\mathbf{s}$, controlling the strength $S = \|\mathbf{s}\|_2$ as well as the direction of the augmentation in $\mathbb{R}^d$. In general, the parameter $\mathbf{s}$, interpreted as a suitably distributed random variable, models the randomness of the transformation. Randomly sampling $\mathbf{s}$, yields a value $\mathbf{s}^{p,k}$ corresponding to a fixed transformation $\mathcal{T}_{\mathbf{s}^{p,k}}$ of strength $S^k$. For instance, for affine translations, $\mathbf{s}^{p,k}$ models a random radial direction sampled from a hypersphere centered at $\mathbf{x}_n$, with strength $S^k$ denoting the magnitude of the translation (e.g. 4-pixel shift). For photometric transformations, $\mathbf{s}^{p,k}$ may model the change in brightness, contrast, hue, and saturation, with total strength $S^k$.

To generate on-manifold paths $\boldsymbol{\pi}_p$ starting from $\mathbf{x}_n$, we proceed as follows. First, we fix a sequence of $K + 1$ strengths $S^0 < S^1 < \ldots < S^K$, with $S^0 = 0$ denoting the identity transformation $\forall\ p$. Then, for each strength $S^k$, with $k \geq 1$, $p$ random directions $\mathbf{s}^{p,k}$ are sampled, each with respective fixed magnitude $\|\mathbf{s}^{p,k}\|_2 = S^k$. This yields $P$ sequences of transformations $\{\mathcal{T}_{\mathbf{s}^{p,k}}\}_{k=0}^K$, each producing augmented versions $\mathbf{x}_n^{p,k}$ of $\mathbf{x}_n$, ordered by strength, $\mathbf{x}_n^{p,1} \prec \ldots \prec \mathbf{x}_n^{p,K}$, and forming a path $\boldsymbol{\pi}_p \subset \mathbb{R}^d$. Specifically, each path $\boldsymbol{\pi}_p$ approximates an on-manifold trajectory by a sequence of Euclidean segments $\mathbf{x}_n^{p,k+1}\mathbf{x}_n^{p,k}$, for $k = 0, \ldots, K$. The maximum augmentation strength $S^K$ controls the distance traveled from $\mathbf{x}_n$, while the number $K$ of strengths used controls how fine-grained the Euclidean approximation is. Pseudocode for generating geodesic paths is presented in section D.

**Volume integration** Once a sequence of paths $\{\boldsymbol{\pi}_p\}_{p=1}^P$ is generated for $\mathbf{x}_n$, volume-based sharpness is computed by integrating over each path $\boldsymbol{\pi}_p$, and normalizing the measure by the length $\text{len}(\boldsymbol{\pi}_p)$ of each path:

$$\frac{1}{P} \sum_{p=1}^P \frac{1}{\text{len}(\boldsymbol{\pi}_p)} \int_{\boldsymbol{\pi}_p} \boldsymbol{\sigma}(\mathbf{x}, y_n) d\mathbf{x} \tag{6}$$

where $\boldsymbol{\sigma}$ represents an infinitesimal sharpness measure, namely the Jacobian and tangent Hessian norms at $(\mathbf{x}_n, y_n)$. The same method can also be applied to accuracy and crossentropy loss to evaluate consistency and confidence of the models predictions over volumes. Figures 1c and 1d illustrate geodesic MC integration. For each training point, $P$ geodesic paths are generated, each anchored to the data manifold by $K$ augmentations. Integrating infinitesimal measures over each path returns a MC sample of sharpness along $\boldsymbol{\pi}_p$. Then, volumetric sharpness is estimated by MC integration over $P$ samples. Importantly, the number $P$ of paths is fixed throughout all experiments, representing the number of MC samples for volume-based integration. Finally, we take a mean-filed view by averaging over the training set $\mathcal{D}$:

$$\frac{1}{P}\mathbb{E}_\mathcal{D} \sum_{p=1}^P \frac{1}{\text{len}(\boldsymbol{\pi}_p)} \int_{\boldsymbol{\pi}_p} \boldsymbol{\sigma}(\mathbf{x}, y_n) d\mathbf{x} = \frac{1}{NP} \sum_{n=1}^N \sum_{p=1}^P \frac{1}{\text{len}(\boldsymbol{\pi}_p)} \int_{\boldsymbol{\pi}_p} \boldsymbol{\sigma}(\mathbf{x}, y_n) d\mathbf{x} \tag{7}$$

Importantly, extending LeJeune et al. (2019), we replace Euclidean integration by geodesic integration over a local approximation of the data manifold, by generating augmentations of increasing strength.

Crucially, the proposed MC integration captures average-case sharpness in proximity of the training data and is directly related to the generalization ability of the studied networks, as opposed to worst-case sensitivity, as typically considered in adversarial settings (Moosavi-Dezfooli et al., 2019). In fact, the random sampling performed in Equation 7 is unlikely to hit adversarial directions, which are commonly identified by searching the input space through an optimization process (Goodfellow et al., 2014; Szegedy et al., 2013).

To conclude our methodology, in section 3.4 we present the family of transformations $\mathcal{T}_\mathbf{s}$ used for generating trajectories $\boldsymbol{\pi}_p$ throughout our experiments.

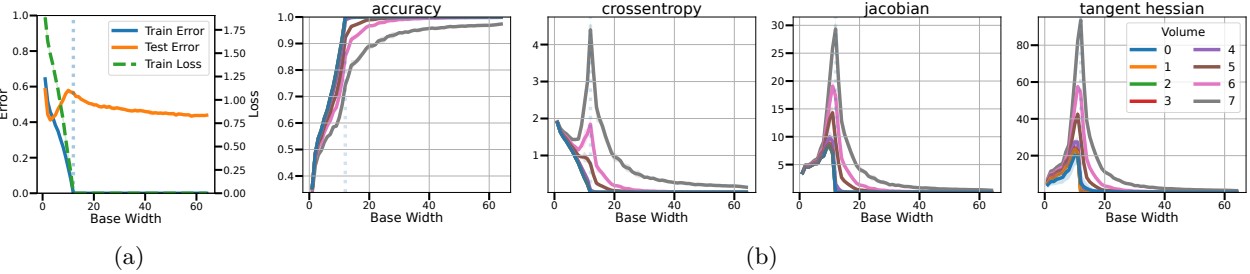

Figure 2: a) Double descent curve for the test error for ConvNets trained on CIFAR-10 with 20% noisy labels. b) Average metrics integrated over volumes of increasing size. Volumes are denoted by the number $K$ of weak augmentations used to generate each geodesic path. From left to right: average training accuracy, training loss, Jacobian norm and Hessian norm, each plotted against model size. The dotted vertical line marks the model width that achieves zero train error (i.e. the *interpolation threshold*). All models are trained for 4k epochs. We observe accuracy over volumes increases monotonically with model size, while crossentropy follows double descent. Combined, the two observations suggest that large networks *confidently* predict the training targets over increasingly large volumes around the training data (for increasing model size). Importantly, interpolation is *sharp* at the interpolation threshold, even infinitesimally at each training point (blue curves), while increasing overparameterization produces *smooth* interpolation, contrary to existing intuition. Shaded areas mark standard deviations over 3 seeds.

### 3.4 Weak Data Augmentation Strategies

Computing sharpness of interpolation via Equation 6 for each data point $\mathbf{x}_n$ requires generating $P$ trajectories $\boldsymbol{\pi}_p$ composed of augmentations of $\mathbf{x}_n$ of controlled increasing strength. Furthermore, the augmented data points $\{\mathbf{x}_n^{p,k}\}_{k=0}^K$ should lie in proximity of the base point $\mathbf{x}_n$ in order for the Euclidean approximation to be meaningful. Finally, to correctly estimate correlation between smoothness and the generalization ability of the networks considered, volume-based sharpness should not rely on validation data points, i.e. the augmentations $\mathbf{x}_n^{p,k}$ should be strongly correlated to $\mathbf{x}_n$, for each $p, k$.

To satisfy the above, we modify a weak data augmentation algorithm introduced by Yu et al. (2018), which allows to efficiently generate augmentations that lie in close proximity to the base training point $\mathbf{x}_n$, for image data. Specifically, each base image $\mathbf{x}_n$, consisting of $C$ input channels (e.g. $C = 3$ for RGB images) and $h \times w$ spatial dimensions, is interpreted as $C$ independent matrices $\mathbf{x}_n[c, :, :] \in \mathbb{R}^{h \times w}$, each factorized using Singular Value Decomposition (SVD), yielding a decomposition $\mathbf{x}_n[c, :, :] = U^c \Sigma^c V^{cT}$, where $\Sigma^c$ is a diagonal matrix whose entries are the singular values of $\mathbf{x}_n[c, :, :]$ sorted by decreasing magnitude. In the original method, Yu et al. (2018) produce weak augmentations by randomly erasing one singular value from the smallest ones, thereby obtaining a modified matrix $\tilde{\Sigma}^c$, and then reconstructing each channel of the base sample via $U^c \tilde{\Sigma}^c V^{cT}$. In this work, in order to generate $P$ random augmentations of strength $k$, $\tilde{\Sigma}^c$ is obtained by erasing $k$ singular values $\Sigma_{i,i}^c$, for $i = w - k - p + 1, \ldots, w - p$, and $p = 0, \ldots, P - 1$[2]. Essentially, the augmentation strength is given by the number $k$ of singular values erased, and $P$ augmentations of similar strength are generated by erasing $P$ subsets of size $k$ from the smallest singular values, for each channel $c$.

We note that this method produces augmented images that are highly correlated with the corresponding base training sample, and as such they do not directly amount to producing validation data points. We refer the reader to appendix E for further details. In the next section, we present our empirical study of sharpness of interpolation for neural networks in relationship to double descent.

## 4 Experiments

In this section, we present our empirical exploration of input-space smoothness of the loss landscape of deep networks as model size and number of training epochs vary. Focusing on *implicit regularization* (Neyshabur

---

[2]Assuming square spatial dimensions $h = w$.

et al., 2015) promoted by optimization and model architecture, we evaluate our sharpness measures on networks with increasing number of parameters, trained without any form of explicit regularization (e.g. weight decay, batch normalization, dropout). We extend our analysis to common training settings in section 4.4.

**Experimental setup**   We reproduce deep double descent by following the experimental setup of Nakkiran et al. (2019b). Specifically, we train a family of ConvNets formed by 4 convolutional stages of controlled base width $[w, 2w, 4w, 8w]$, for $w = 1, \ldots, 64$, on the CIFAR-10 dataset with 20% noisy training labels and on CIFAR-100. All models are trained for 4k epochs using SGD with momentum 0.9 and fixed learning rate. Following Arpit et al. (2019), to stabilize prolonged training, we use a learning rate warmup schedule. Furthermore, we extend our empirical results to training settings more commonly found in practice, and validate our main findings on a series of ResNet18s (He et al., 2015) of increasing base width $w = 1, \ldots, 64$, with batch normalization, trained with the Adam optimizer for 4k epochs using data augmentation. We refer the reader to section B for a full description of our experimental setting. In section G.1, we extend our main results to Transformer networks trained on machine translation tasks.

We begin our experiments by reproducing double descent for the test error for the ConvNets (Figure 2a). Starting with small models and by increasing model size, a U-shaped curve is observed whereupon small models underfit the training data, as indicated by high train and test error. As model size increases, the optimal bias/variance trade-off is reached (Geman et al., 1992). Mid-sized models increasingly overfit training data – as shown by increasing test error for decreasing train error and loss – until zero training error is achieved, and the training data is interpolated. The smallest interpolating model size is typically referred to as *interpolation threshold* (Belkin et al., 2019). Near said threshold, the test error peaks. Finally, large overparameterized models achieve improved generalization, as marked by decreasing test error, while still interpolating the training set.

## 4.1   Loss Landscape Smoothness Follows Double Descent

In this section, we establish a strong correlation between double descent of the test error and smooth interpolation of noisy training data. Figure 2b studies fitting of training data for models at convergence (training for 4k epochs) as model size increases. Starting with (infinitesimal) sharpness at training points (blue curve), we observe that training accuracy at convergence monotonically increases with model size, with 100% accuracy reached at the interpolation threshold and maintained therefrom. At the same time, crossentropy loss over volumes follows double descent, with peak near the interpolation threshold, and then decreasing as model size grows. Similarly, the Jacobian and Hessian norms peak at the interpolation threshold and then rapidly decrease, showing that all training points become stationary for the loss, and that the landscape becomes flatter as model size grows past the interpolation threshold. When all measures are integrated over volumes of increasing size (number $K$ of augmentations per path), we observe how large overparameterized models are able to smoothly fit the training data over large volumes. This finding suggests that – in contrast to the polynomial intuition of Figure 1a) – overparameterized networks interpolate training data *smoothly* (as intuitively depicted in Figure 1b).

Our finding extends the observations of Novak et al. (2018) and LeJeune et al. (2019) from fixed-size networks to a spectrum of model sizes, and establishes a clear correlation with the test error peak in double descent. Finally, the results substantiate the universal law of robustness (Bubeck & Sellke, 2021), showing that at the interpolation threshold highest sensitivity to input perturbations is observed, while overparameterization beyond the threshold promotes smoothness. Intriguingly, our findings represent mean sharpness as opposed to the worst case studied by Bubeck & Sellke (2021), showing that the observed regularity is much stronger in practice. In the following section, we study this behaviour in proximity of cleanly- and noisily-labeled training samples. We refer the reader to section 4.4 for analogous results on ResNets trained with Adam.

## 4.2   Smooth Interpolation of Noisy Labels

In this section, we break down the noisily labeled training set into two subsets: cleanly-labeled points, and training points with corrupted labels, and explore how fitting is affected by the training labels.

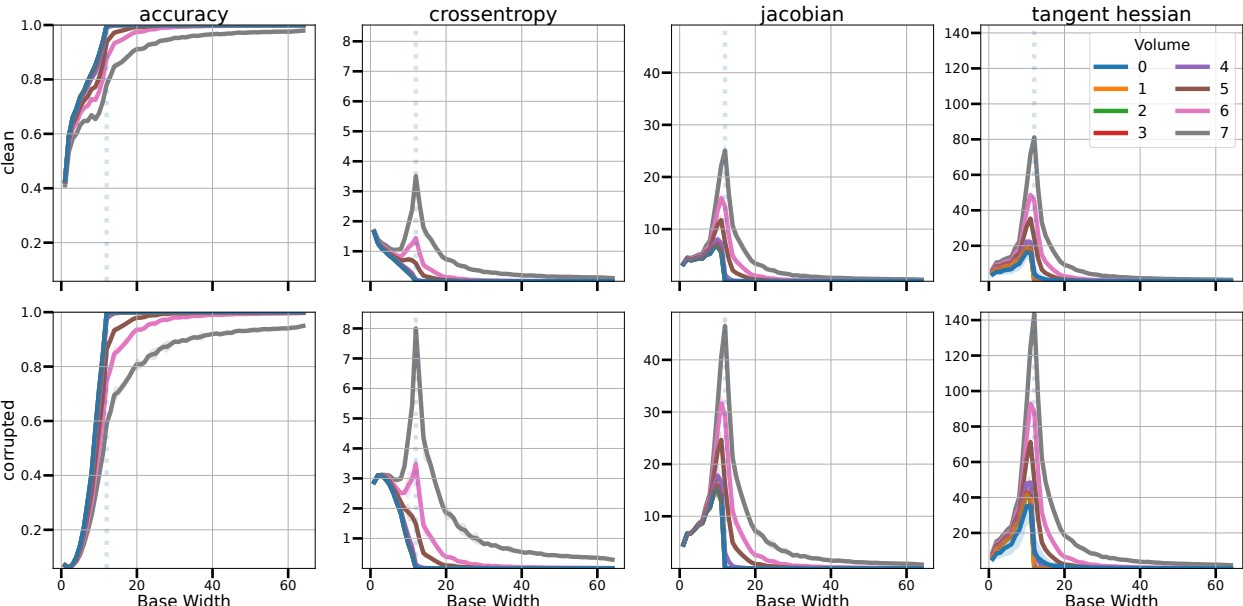

Figure 3: Average accuracy, crossentropy, Jacobian and Hessian norms integrated over volumes of increasing size (augmentations per path) around clean (top) and noisy (bottom) subsets of the CIFAR-10 training set with 20% noisy labels. For models near the interpolation threshold, we observe a large increase in the loss for increasing neighborhood size. At the interpolation threshold, sharp interpolation is observed for both clean and noisy samples, with crossentropy, sensitivity (Jacobian norm) and curvature peaking over all volumes considered. Larger models present a smoother loss landscape around training points, with the largest models expressing a locally flat landscape around each point. This finding shows that large networks are confidently and smoothly *predicting the noisy labels* around data points whose label was corrupted, suggesting that smoothness emerging from overparameterization in fact hinders generalization locally to those points.

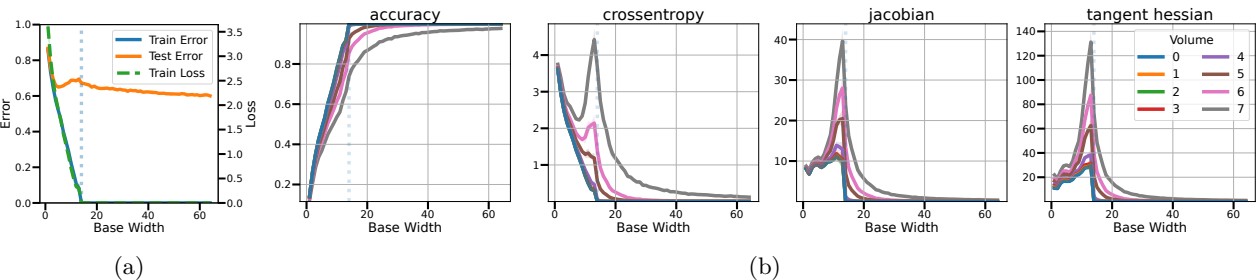

Figure 4: a) Double descent for the test error for ConvNets trained on CIFAR-100. b) Average metrics integrated over volumes of increasing size (number $K$ of augmentations per path). From left to right: average training accuracy, crossentropy, Jacobian, and Hessian norm, each plotted against model size. All models are trained for 4k epochs. For relatively complex datasets (i.e. with few samples per class), our findings hold even without artificially corrupted labels, suggesting that the trends reported in this work are not caused by synthetic noise. Shaded areas depict standard deviations over 3 seeds.

Figure 3 reports accuracy, crossentropy, as well as sharpness measures computed on the clean subset of CIFAR-10 (top), as well as the corrupted subset (bottom), for volumes of increasing size. We begin by noting that small models fit mostly the cleanly labeled data points, and show close to zero accuracy on the noisily labeled data points, showing a bias towards learning simple patterns. We hypothesize that most cleanly labeled samples act as "simple examples", while noisily labeled ones provide "hard examples", akin to support vectors, for small size models. This behaviour is aligned with prior observations, reporting that

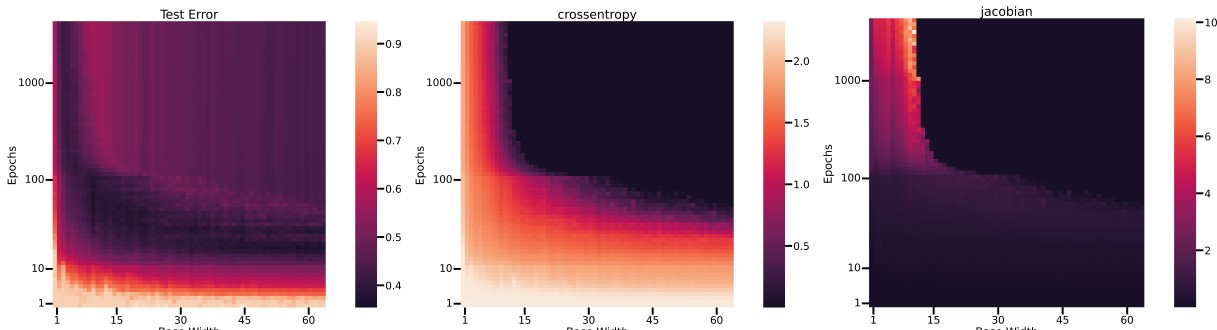

Figure 5: (Left) Test error (Middle) Train crossentropy (Right) Jacobian norm for ConvNets trained on CIFAR-10 with 20% noisy labels. The heatmaps show each metric for increasing training epochs (y-axis) and base width (x-axis). Models past the interpolation threshold (base width $w = 15$) undergo epoch-wise double descent for each metric. Similar trends are observed for curvature, as measured by the Hessian norm. All metrics are computed on the training set only, without geodesic Monte Carlo integration.

deep networks admit support vectors (Toneva et al., 2018) and that deep networks share the order in which samples are fitted (Hacohen et al., 2020). We refer the reader to Hacohen et al. (2020) for details.

As model size grows toward the interpolation threshold, networks fit both clean and noisy samples (as marked by increasing accuracy on both subsets), with large models consistently predicting the clean and noisy labels over large volumes. At the same time, crossentropy local to each training point (blue curve) approaches zero past the interpolation threshold, while volume-based crossentropy undergoes double descent. Interestingly, this trend is observed both around cleanly- and noisily-labeled training samples, with peaks at the interpolation threshold which are considerably more marked for noisy labels.

Our sharpness measures follow double descent for all volumes considered, even when no Monte Carlo integration is performed (blue curve). Importantly, curvature as measured by the Hessian norm rapidly decreases as model size grows, showing that large networks smoothly interpolate both clean and noisy samples. Importantly, we observe how the second descent in test error corresponds to improved fitting of cleanly-labeled samples, while the network lose their generalization ability locally to noisy labeled points.

In Figure 4a we extend the observations to CIFAR-100, where model-wise double descent is observed even on the standard dataset without artificially corrupted labels. Similarly to what observed on CIFAR-10, Figure 4b shows the loss landscape peaking in sharpness at the interpolation threshold, and then rapidly decreasing as model size grows, with large networks smoothly fitting the training set over increasingly large volumes. This finding suggests that double descent is tied to dataset complexity, and that the trends reported in this work are not caused by artificially corrupted labels.

### 4.3 Epochwise Double-Descent

We now turn our attention to epoch-wise double descent, first reported for the test error of deep networks by (Nakkiran et al., 2019b). Figure 5 shows the test error (left), train crossentropy (middle), as well as Jacobian norm (right) for ConvNets trained on CIFAR-10 with 20% noisy labels. We consolidate our observations for each metric with heatmaps, in which the y-axis represents training epochs, and the x-axis denotes the models' base width. We observe that models past the interpolation threshold (base width $w = 15$) undergo epoch-wise double descent for each metric. At the same time, models with base width $w < 15$ are unable to reduce their test error within 4k training epochs, and this is associated to non-decreasing training loss as well as Jacobian norm. We hypothesize that the model size affects a model's ability to interpolate the training data, and therefore affects the training dynamics and the occurrence of epoch-wise double descent.

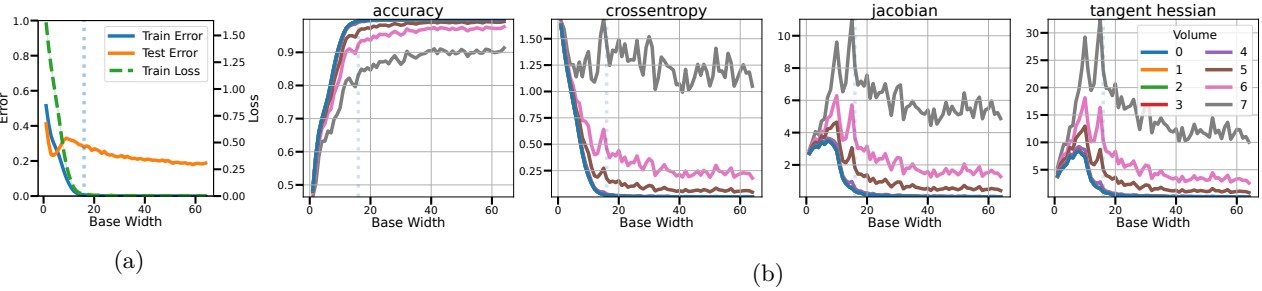

(a)                                                  (b)

Figure 6: (a) Double descent for the test error the CIFAR-10 with 20% noisy labels for a family of ResNet18s of increasing base width $w$, trained with data augmentation. (b) Accuracy, crossentropy, Jacobian and Hessian norms over volumes. All models are trained for 4k epochs. Analogous trends as observed for the ConvNets holds in this case. However, the largest integration volume considered ($K = 7$ augmentations per path), now shows considerably increased sharpness and loss curvature, while still undergoing double descent.

## 4.4   Practical Training Settings

So far, the training setting included the least amount of confounders (e.g. adaptive learning rates, explicit regularization, skip connections, normalization layers) and focused on implicit regularization. In Figure 6, we extend our findings to ResNets trained on CIFAR-10 with 20% noisy labels, with the Adam optimizer, data augmentation (4-pixel shifts and random horizontal flips), as well as batch normalization layer (see appendix B for details). Both model-wise and epoch-wise trends reported for the ConvNets also hold for this setup, with the interpolation threshold occurring at base width $w = 18$. We consolidate our model-wise and epoch-wise findings with heatmaps in Figure 12 and 14. Interestingly, data augmentation causes the peak in test error to occur earlier than the interpolation threshold. We hypothesize that the mismatch – which can also be observed in related works (Nakkiran et al., 2019b) – is due to a lack of fine grained control over model size as base width $w$ varies. Importantly, for large volumes around training points ($K = 7$ augmentations per path), training accuracy degrades and loss sharpness increases. However, all sharpness metrics undergo double descent as model size grows, confirming the trends reported in simpler training settings.

## 4.5   Towards Decoupling Smoothness from Generalization

Our experimental results suggest that double descent in the test error is closely related to input-space smoothness. One possible interpretation is that models at the interpolation threshold learn small and irregular decision regions, marked by high loss sharpness, while large models learn more regular decision regions with wider margins, supporting the observations of Jiang et al. (2019).

In fact, as consistently observed in our experiments, on the one hand, models near the interpolation threshold fail to smoothly interpolate all clean samples, while on the other hand large models can smoothly interpolate the entire training set. This effectively enforces a trade-off for which large models lose generalization ability around noisy samples, but can correctly classify all clean samples. Assuming the train and test distributions are similar (i.e. excluding covariate shifts), this would in turn result in improved average test error past the interpolation threshold, as indeed observed in practice. To assess the validity of our interpretation, we decouple smoothness from generalization by studying training settings in which smooth training set interpolation hurts generalization. In this setting, we expect smooth interpolation to consistently emerge with overparameterization, but this time without producing double descent in the test error. To corroborate our interpretation, in principle, one would need to construct a nearest neighbour classifier (either in input or in feature space), and test whether predictions for each test sample are affected by proximity to corrupted samples. In the following, we propose a simple experiment to decouple smoothness from generalization, without requiring knowledge of proximity of test samples to train samples.

First, we corrupt 20% of the CIFAR-100 training set with asymmetric label noise, such that 80% samples of 20 randomly selected classes are perturbed. At test time, this enables us to split the test set into (1) samples whose classes have been corrupted, and (2) samples belonging to unperturbed classes. Figure 7a

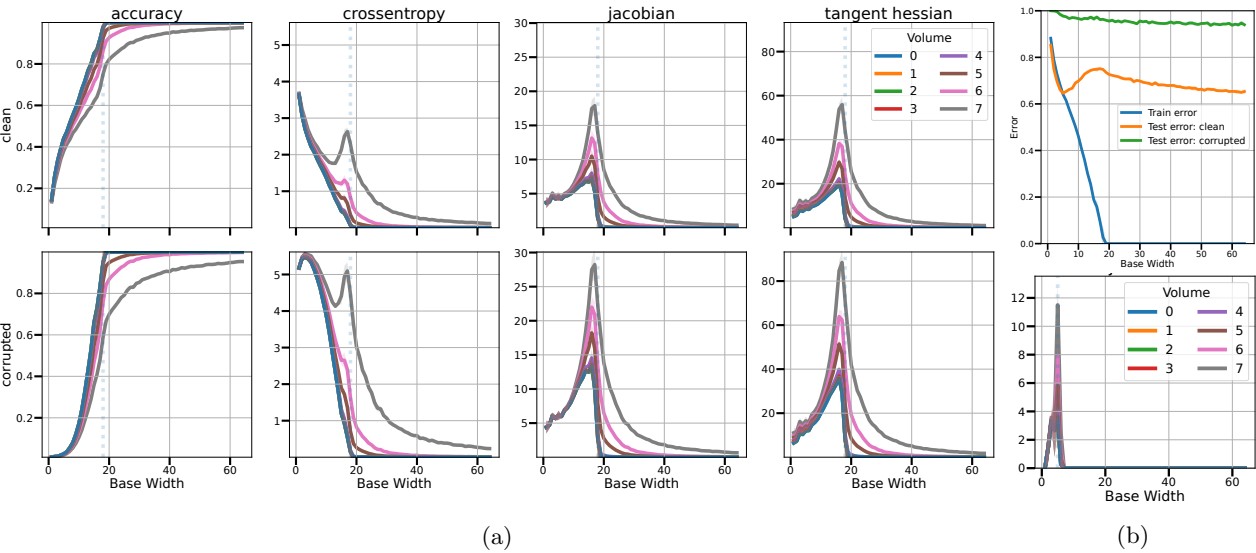

Figure 7: **Decoupling smoothness from generalization**. We present an experiment in which 20 randomly selected classes of the CIFAR-100 training split are corrupted with asymmetric label noise, perturbing 80% training labels within each class, for a total of 20% corrupted training samples. At test time, this enables splitting the test set into classes that have been corrupted at train time, and unperturbed classes. (a) Overparameterization promotes a smooth and flat loss landscape around both cleanly-labeled as well as noisy training samples under asymmetric label noise. (b, top) Confirming our hypothesis, double descent for the test error can still be observed for the unperturbed classes, while the trend disappears for the corrupted classes. This finding shows that overparameterization promotes smoothness in the input variable, which is aligned with generalization only around cleanly labeled points. (b, bottom) For networks trained on 100% noisy labels, smooth interpolation still follows double descent over volumes of increasing size around training points, but such property in this case is not aligned with generalization.

shows that, even under strong asymmetric noise, overparameterization promotes input-space smoothness over increasingly large volumes around both clean and noisy training samples. Perhaps surprisingly, in Figure 7b (top) at test time double descent is still observed for test samples belonging to unperturbed classes, while the trend disappears for the corrupted classes. This confirms our interpretation and shows that double descent should be understood in terms of input-space smoothness, and its relation to generalization.

Second, we train ResNets on CIFAR-10 with *all training labels corrupted* (Figure 7b, bottom). Also in this setting, loss sharpness over volumes follows double descent, peaking near the interpolation threshold, and decreasing with increasing model size. Trivially, all networks in this setting lose their generalization ability, with performance close to random chance. This finding shows that overparameterization promotes smooth interpolation of the training data, and that such property is not necessarily aligned with generalization.

## 5 Conclusions

In this work, we present geodesic Monte Carlo integration tools to study the input space of neural networks, providing intuition – built on extensive experiments – on how neural networks fit training data. We present a strong correlation between epoch-wise as well as model-wise double descent for the test error and smoothness of the loss landscape in input space. Our experiments show that overparameterization promotes input space regularity via smooth interpolation of clean and noisy training data, which is aligned with improved generalization for datasets with relatively low ratio of label noise. Crucially, contrary to intuitions in polynomial regression, deep networks uniformly predict noisy training targets over volumes around noisily-labeled training samples – a behaviour which may have severe negative impact in practical applications with imbalanced training sets or with covariate shifts of the population distribution.

Consistently in our experiments, we observe a peak in test error and loss sharpness near the interpolation threshold, which decreases for better generalizing models. Finally, for increasing volumes around each training point, we observe that overparametrization promotes flatter minima of the loss *in input space*, providing initial clues as to why large overparameterized models generalize better, and corroborating the findings of Somepalli et al. (2022) on regularity of decision boundaries of overparameterized classifiers, as well as Gamba et al. (2022) on input-space regularity.

Our analysis substantiates the law of robustness of Bubeck & Sellke (2021), and extends the findings of Novak et al. (2018) to experimental settings with controlled model size. We hypothesize that overparameterization affects the dynamics of optimization and interpolation, promoting a smooth loss landscape. An interesting open problem is characterizing the impact of individual layers on interpolation, as model size grows.

Finally, our analysis opens the question of whether increased interpolation smoothness is to be attributed to the model architecture, the optimizer, or a combination of both. First, increased network width, as controlled in our experiments, has been recently connected to the existence of paths connecting critical points for the optimizer (Simsek et al., 2021), suggesting that model width plays an important role in affecting the dynamics of the optimizer. Particularly, one or mode connected manifold of minima may allow wider networks to retain interpolation while at the same time optimizing for input-space smoothness (Li et al., 2021). Second, understanding the existence of implicit regularization promoted by the optimizer is at present an active area of research. On the one hand, several studies argue that stochastic optimization, and the potential implicit regularization effect of mini-batch noise, are not required for generalization (Chiang et al., 2023; Paquette et al., 2022; Geiping et al., 2020). On the other hand, current models of double descent hypothesize that stochastic noise is an important component in explaining implicit regularization and double descent in deep learning (Li et al., 2021; Blanc et al., 2020).

### Acknowledgments

This work was partially supported by the Wallenberg AI, Autonomous Systems and Software Program (WASP) funded by the Knut and Alice Wallenberg Foundation. Scientific computation was enabled by the supercomputing resource Berzelius provided by National Supercomputer Centre at Linköping University and the Knut and Alice Wallenberg foundation. The work was partially funded by Swedish Research Council project 2017-04609.

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

## A   Appendix

Section B summarizes our experimental setup, while Section C, D, and E respectively detail the tangent Hessian computation method, the geodesic path generation algorithm, and the weak data augmentation strategy used for geodesic Monte Carlo integration. Finally, in section F we extend our discussion of related works. Additional experiments are reported in section G.

## B   Network Architectures and Training Setup

**Network Architectures**   The ConvNets and ResNets used follow the experimental settings of Nakkiran et al. (2019b), with the only difference that we disable batch normalization in order to focus our study on implicit regularization. In summary, the ConvNets are composed of 4 convolutional stages (each with a single conv + ReLU block) with kernel size $3 \times 3$, stride 1, padding 1, each followed by maxpooling of stride 2 and kernel size $2 \times 2$. Finally, a max pooling layer of stride 2 and kernel size $2 \times 2$ is applied, followed by a linear layer. The Residual networks used in this study are ResNet18s (He et al., 2015) without batch normalization.

Both ConvNets and ResNets are formed by 4 convolutional stages at which the number of learned feature maps doubles, i.e. the base width of each stage follows the progression $[w, 2w, 4w, 8w]$, with $w = 64$ denoting a standard ResNet18. To control the number of parameters in each network, the base width $w$ varies from 1 to 64.

Throughout our experiments, augmentations $\tilde{\mathbf{x}}_n$ of a sample $(\mathbf{x}_n, y_n)$ are labelled with their respective (potentially noisy) training target $y_n$.

**Dataset Splits**   To tune the training hyperparameters of all networks, a validation split of 1000 samples was drawn uniformly at random from the training split of CIFAR-10 and CIFAR-100.

**ConvNet Training Setup**   The training settings are the same for CIFAR-10 and CIFAR-100. All ConvNets are trained for 4k epochs with SGD with momentum 0.9, fixed learning rate $1e-3$, batch size 128, and no weight decay. All learned layers are initialized with Pytorch's default weight initialization (version 1.11.0). To stabilize prolonged training in the absence of batch normalization, we use learning rate warmup: starting from a base value of $1e-4$ the learning rate is linearly increased to $1e-3$ during the first 5 epochs of training, after which it remains constant at $1e-3$.

**ResNet Training Setup**   All ResNets are trained for 4k epochs using Adam with base learning rate $1e-4$, batch size 128, and no weight decay. All learned layers are initialized with Pytorch's default initialization (version 1.11.0). All residual networks are trained with data augmentation, consisting of $4 - pixel$ random shifts, and random horizontal flips.

**Computational Resources**   Our experiments are conducted on a local cluster equipped with NVIDIA Tesla $A$100s with 40GB onboard memory. For each dataset and architecture, we train 64 different networks for 4000 epochs with 3 different seeds. The total time for computing our experiments, excluding training networks and hyperparameter finetuning, amounts to approximately 6 GPU years. Furthermore, computing our statistics requires evaluating per-sample Jacobians for each training point and corresponding augmentations, for increasing volumes around each point. For each training setting, this was performed for 72 model checkpoints collected during training, to produce the heatmaps in Figures 5, 11, 12, 13 and 14.

## C   Tangent Hessian Computation

To estimate the tangent Hessian norm at a point $\mathbf{x}_n$ through Equation 5, we approximate the tangent space to the data manifold local to $\mathbf{x}_n$ by using a set of random weak augmentations of $\mathbf{x}_n$. To guarantee that all augmentations $\mathbf{x}_n + \mathbf{u}_m$, as well as the displacements $\mathbf{x}_n + \delta \mathbf{u}_m$ lie on the data manifold, we use weak colour augmentations as follows.

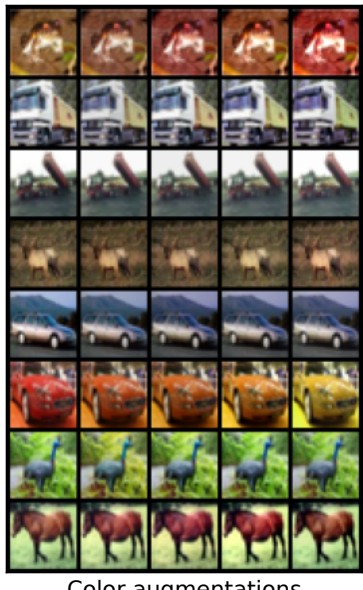

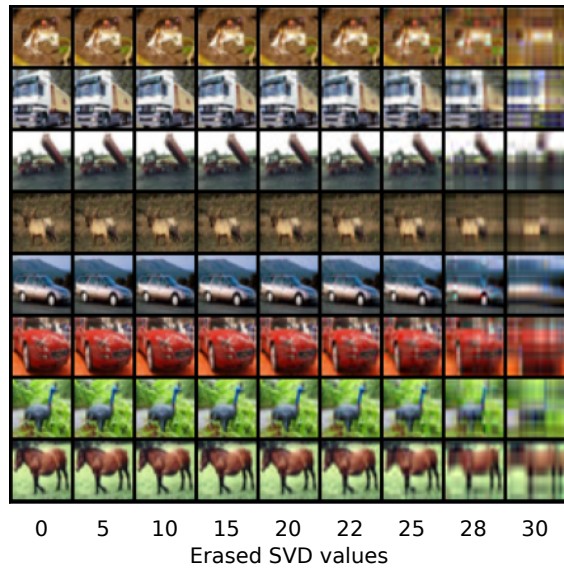

Color augmentations

0   5   10   15   20   22   25   28   30

Erased SVD values

Figure 8: (Left) Visualization of random colour augmentations used to estimate the tangent Hessian norm. Each row represents a set of random augmentation, with the first image per-row showing the corresponding base sample. (Right) Each row represents SVD augmentations of increasing strength. Also in this case, the first column represents the base sample used to generate the corresponding augmentations in each row.

For each sample $\mathbf{x}_n$, we apply in random order the following photometric transformations:

- random brightness transformation in the range $[0.9, 1.1]$, with 1. denoting the identity transformation.

- random contrast transformation in $[0.9, 1.1]$, with 1. denoting the identity transformation.

- random saturation transformation in $[0.9, 1.1]$, with 1. denoting the identity transformation.

- random hue transformation in $[-0.05, 0.05]$, with 0. denoting the identity transformation.

Furthermore, a step size $\delta = 0.1$ is used for computing the finite differences in Equation 5. 4 augmentations $\mathbf{x}_n + \mathbf{u}_m$ are sampled for each point. All randomness is controlled to ensure reproducibility. Figure 8 (left) shows a visualization of the colour augmentations used.

## D   Geodesic Paths Generation

In this section, we provide pseudocode for the algorithm used for generating geodesic paths, used for Monte Carlo integration. Let $\mathbf{x}_0 \in \mathbb{R}^d$ denote a training point, which we use as the starting point of geodesic paths $\boldsymbol{\pi}_p$ emanating from $\mathbf{x}_0$. Let $\mathcal{T}_{\mathbf{s}} : \mathbb{R}^d \to \mathbb{R}^d$ denote a family of smooth transformations (data augmentation), dependent on a parameter $\mathbf{s}$ controlling the magnitude and direction of the transformation (e.g. radial direction and displacement for pixel shifts). Let $\mathcal{S} := \{\mathbf{s}^1, \ldots, \mathbf{s}^K\}$ denote a sequence of parameters for the family $\mathcal{T}_{\mathbf{s}}$, each with strength $S^k = \|\mathbf{s}^k\|_2$ for $k = 1, \ldots, K$, such that $S^1 < \ldots < S^K$. Then, Algorithm 1 returns a geodesic path $\boldsymbol{\pi} : [0, 1] \to \mathbb{R}^d$, based at $\mathbf{x}_0$, i.e. $\boldsymbol{\pi}(0) = \mathbf{x}_0$, which is anchored to the data manifold local to $\mathbf{x}_0$ by a sequence of augmentations of increasing strength, for $k = 1, \ldots, K$.

Particularly, Algorithm 1 can be applied $P$ times to generate paths $\boldsymbol{\pi}_p$ emanating from $\mathbf{x}_0$. Finally, by integrating metrics of interest (e.g. Jacobian and tangent Hessian norms) along each path $\boldsymbol{\pi}_p$, we obtain

---

**Algorithm 1** Generate a geodesic path $\pi$ emanating from a training point $\mathbf{x}_0$.

---

1: **function** GEODESIC PATH($\mathbf{x}_0$, $\mathcal{T}_{\mathbf{s}}$, $\mathcal{S} := \{\mathbf{s}^1, \ldots, \mathbf{s}^K\}$)
2:     $\mathcal{P} \leftarrow \{\mathbf{x}_0\}$                                          ▷ Set of on-manifold points.
3:     **for** $\mathbf{s}^k \in \mathcal{S}$ **do**
4:         sample $\mathbf{s} \sim \mathbf{s}^k$                          ▷ Sample augmentation of strength $S^k = \|\mathbf{s}\|_2$.
5:         $\mathbf{x}^k = \mathcal{T}_{\mathbf{s}}(\mathbf{x}_0)$                          ▷ Generate weak data augmentation.
6:         $\mathcal{P} \leftarrow \mathcal{P} \cup \{\mathbf{x}^k\}$
7:     **end for**
8:     **return** $\mathcal{P}$                          ▷ Set of data augmentations forming a path
                                          $\pi$, with points sorted by distance from $\mathbf{x}_0$.
9: **end function**

---

estimates of sharpness of the loss along each path, which we use as Monte Carlo samples in Equation 7 for estimating volume-sharpness. We recall that the size of the volume considered is controlled by the maximum augmentation strength $S^K$ used for generating weak augmentations, which is proportional to the distance travelled away from $\mathbf{x}_0$ in input space.

## E  SVD Augmentation

The SVD augmentation method presented in section 3.4 allows for generating images that lie in close proximity to the base sample $\mathbf{x}_n$. Figure 8 shows an illustration of the original image (first column) and several augmented images, as the augmentation strength (number of erased singular values) increases. Figure 9 shows the average (over the CIFAR-10 training set) Euclidean distance of augmented samples from their respective base sample, as well as the length of the polygonal path formed by connecting augmentations of increasing strength. We note that for $k < 30$, in expectation, augmentations lie in close proximity to the original base sample in Euclidean space.

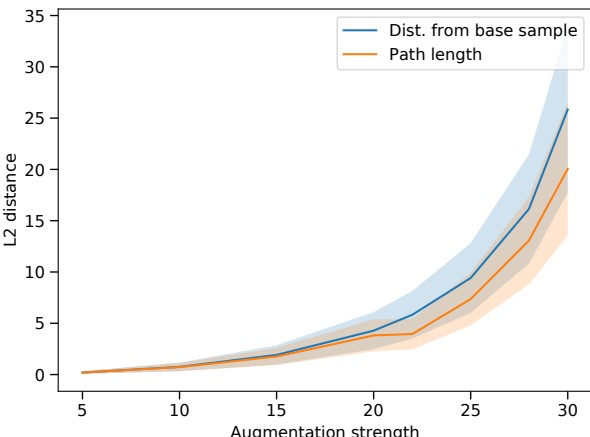

Figure 9: Average L2 distance from the base samples, for augmentations of increasing strength.

## F  Extended Related Works

In this section, we extend the related work discussion of section 2 to contextualize our findings in relationship to linear models.

In linear regression, the model-wise double descent phenomenon has been studied in terms of *harmless interpolation* (Muthukumar et al., 2020) or *benign overfitting* of noisy data (Bartlett et al., 2020), by controlling the number of input features $\boldsymbol{\theta}$ considered. Particularly, for the least squares solution to a noisy linear

regression problem with random input and features, the impact of noise on generalization is mitigated by the abundance of weak features (Belkin et al., 2020). In this context, interpolation is studied for data whose population is described by noisy observations from a linear ground truth model. In the following, we delineate the main differences between linear regression and the experimental setting considered in our study.

We begin by noting that, since the model function of linear models has zero curvature (both w.r.t. model input $\mathbf{x}$ and parameters $\boldsymbol{\theta}$), the only source of nonlinearity and curvature in linear regression is the error function (MSE). To see this, let $f(\mathbf{x}, \boldsymbol{\theta}) = \boldsymbol{\theta}^T \mathbf{x}$ denote a linear regression model, estimated by minimizing the mean squared error $\mathcal{L}(\boldsymbol{\theta}, \mathbf{x}, y) = \frac{1}{2N} \sum_{n=1}^{N} (f(\mathbf{x}_n, \boldsymbol{\theta}) - y_n)^2$, where $y_n$ is a noisy target $\forall n$. Then, the error function $\mathcal{L}$ has constant curvature $H = \|\frac{\partial^2 \mathcal{L}}{\partial \mathbf{x} \partial \mathbf{x}^T}\|_2 = \|\boldsymbol{\theta}\boldsymbol{\theta}^T\|_2$, independent of $\mathbf{x}$.

In contrast, we study the case of nonlinear classification problems and nonlinear models, which have notable differences from the linear case. First, there is no a priori closed form solution of the learning problem, thus providing relevance to empirical studies. Second, curvature of the model function is non-constant, and the function may oscillate arbitrarily outside of the training data (this is known as the Runge phenomenon). Third, studies that rely exclusively on the test error suggest that interpolation is harmless also in overparameterized nonlinear models. Finally, the model function of convolutional architectures is independent of input-data dimensionality, and the relationship between complexity of the model function and its underlying parameterization is therefore implicit.

In this setting, we experimentally show that, in the interpolating regime, (1) curvature at training points depends non-monotonically on model size; (2) oscillations occur especially for small interpolating models, which are worst affected by noise; (3) large models achieve low-curvature interpolation of both clean and noisy samples (in contrast with the polynomial intuition), and such property is observed over large volumes (non-zero measure) around each training point (in contrast with the Runge phenomenon, thus providing evidence of implicit regularization); (4) Interpolation of noise impacts generalization even for large models (contrary to the overparameterized linear regression case); (5) Double descent observed for input space curvature occurs even when fitting 100% noisy data, more clearly pinpointing properties that are consistently promoted by overparameterization in deep nonlinear networks.

Our methodology enables the study of sharpness of fit of training data for nonlinear models, providing a comparative study of the regularity with which different parameterizations achieve interpolation and (in some cases) generalization.

## G    Additional Experiments

### G.1    Transformers

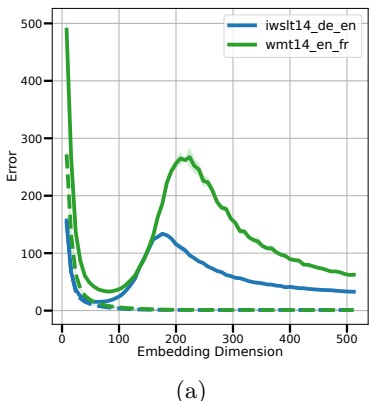
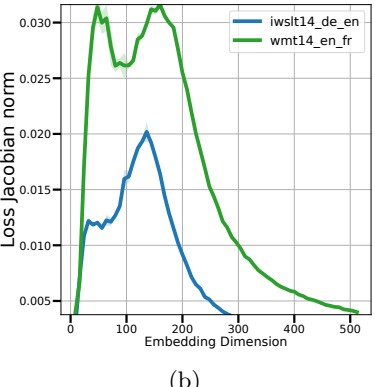

(a)                                      (b)

Figure 10: a) **Double descent** of the test error for transformers trained on translation tasks, as the embedding dimension and model width vary. b) **Average Jacobian norm**.

We consider multi-head attention-based Transformers (Vaswani et al., 2017) for neural machine translation tasks. We vary model size by controlling the embedding dimension $d_e$, as well as the width $h$ of all fully connected layers, which we set to $h = 4d_e$ following the architecture described in Vaswani et al. (2017). We train the transformer networks on the WMT'14 En-Fr task (Macháček & Bojar, 2014), as well as ISWLT'14 De-En (Cettolo et al., 2012). The training set of WMT'14 is reduced by randomly sampling 200k sentences, fixed for all models. The networks are trained for 80k gradient steps, to optimize per-token perplexity, with 10% label smoothing, and no dropout, gradient clipping or weight decay.

For both datasets, Figure 10a shows the double descent curve for the test error for both datasets considered. Figure 10b extends our main result beyond vision models, showing that loss sharpness at each training point, as measured by the Jacobian norm, follows double descent for the test error.

### G.2 ConvNets

Figures 11 and 13 summarize our main findings with heatmaps showing modelwise and epochwise trends for the test error, train loss, as well as our sharpness metrics, individually computed over the clean and noisy subsets of CIFAR-10.

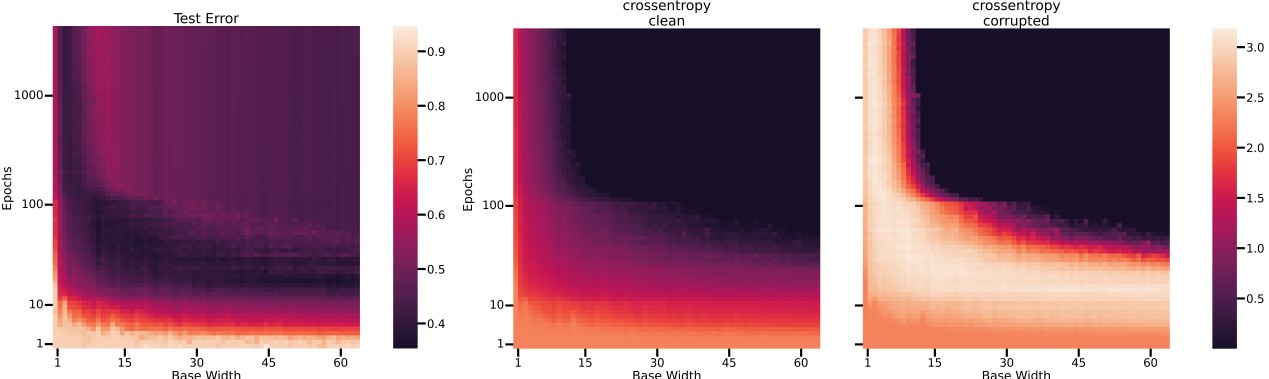

Figure 11: Test error (left), crossentropy loss over cleanly-labelled training samples (middle) and corrupted training samples (right) over epochs (y-axis) for different model sizes (x-axis), for ConvNets on CIFAR-10.

### G.3 ResNets

Figures 12 and 14 present heatmaps showing modelwise and epochwise trends for the test error, train loss, as well as our sharpness metrics, individually computed over the clean and noisy subsets of CIFAR-10.

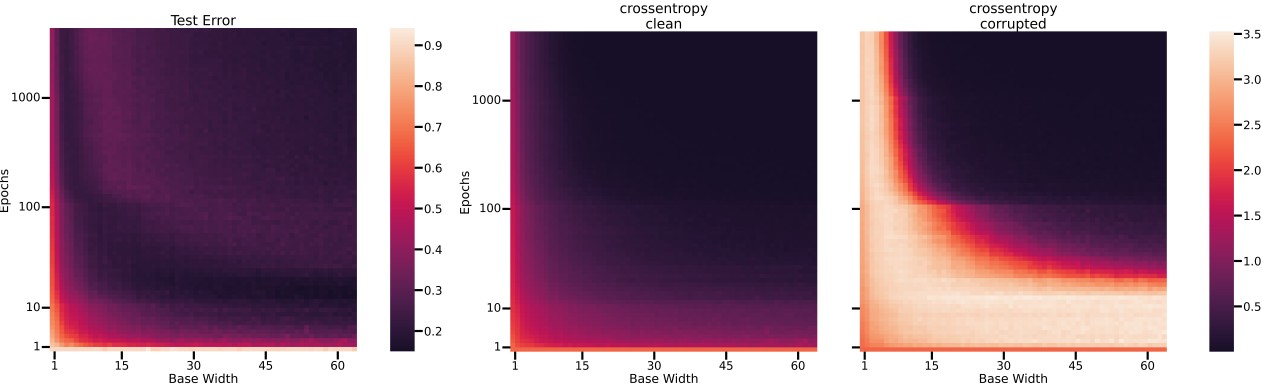

Figure 12: Test error (left), crossentropy loss over cleanly-labelled training samples (middle) and corrupted training samples (right) over epochs (y-axis) for different model sizes (x-axis), for ResNets on CIFAR-10.

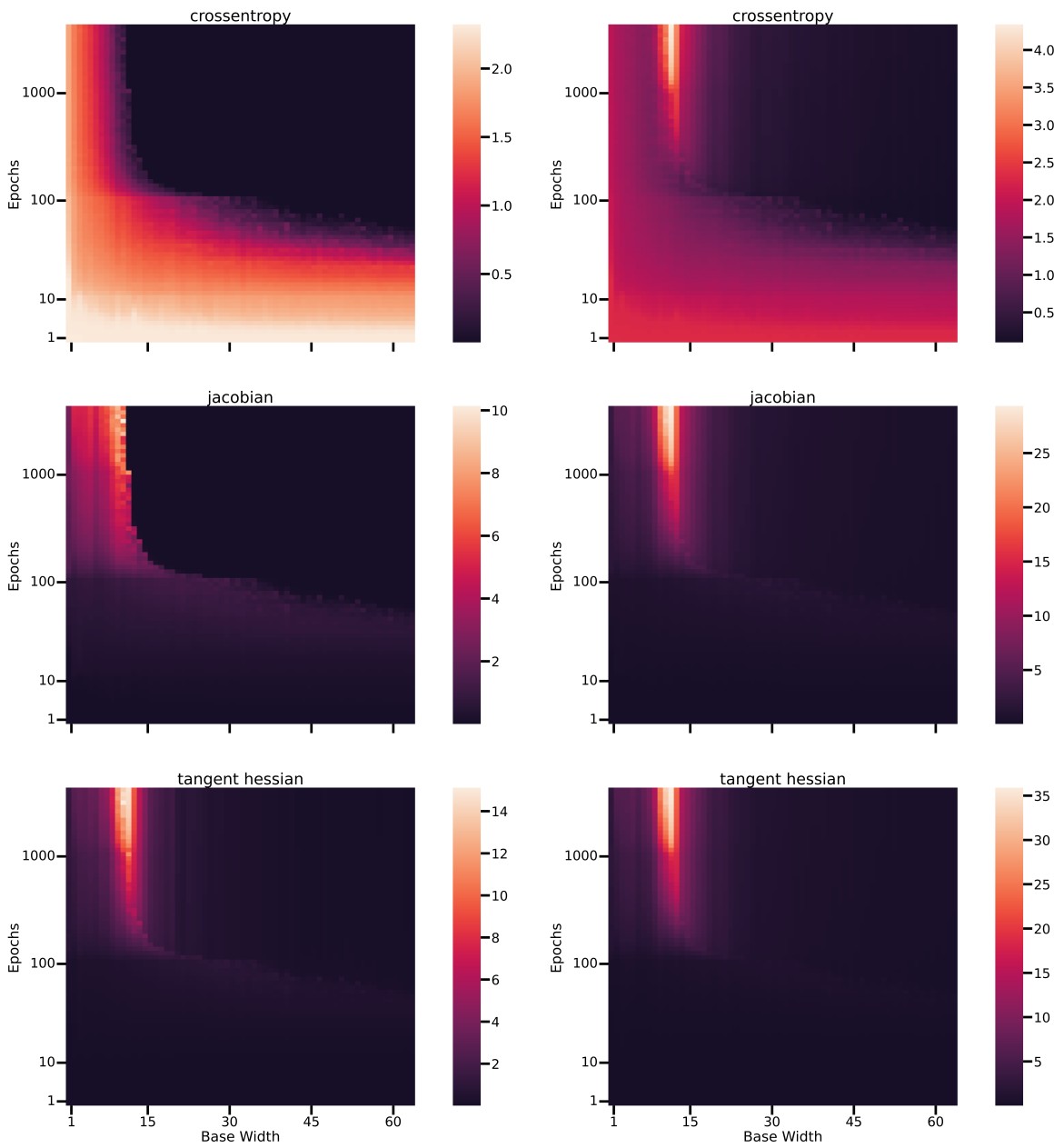

Figure 13: (Left column) Metrics evaluated on the training set without Monte Carlo integration on ConvNets. (Right column) Monte Carlo integration over a neighborhood with paths consisting of 7 augmentations.

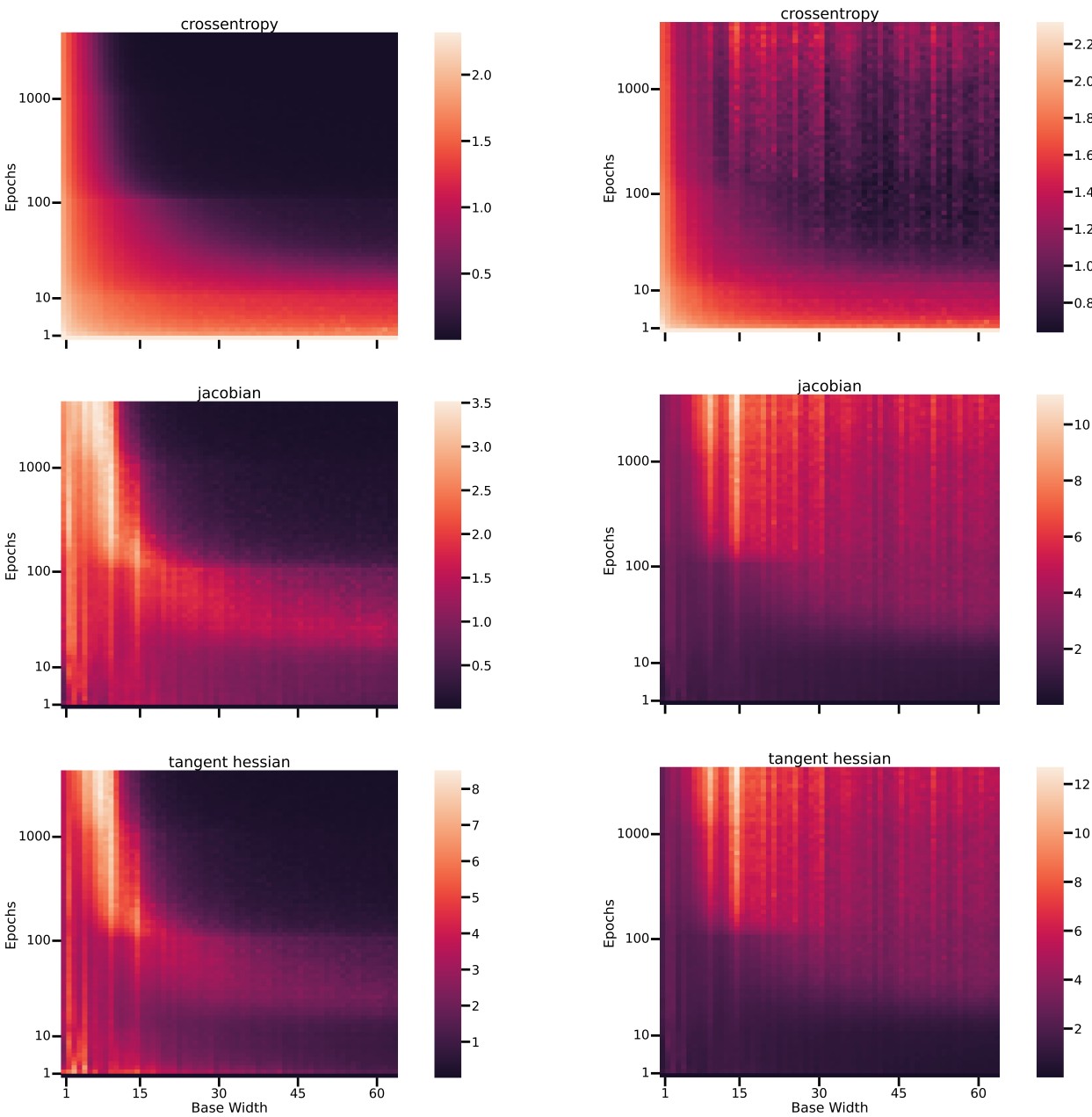

Figure 14: (Left column) Metrics evaluated on the training set without Monte Carlo integration on ResNets. (Right column) Monte Carlo integration over a neighborhood with paths consisting of 7 augmentations.

