# OpenReview forum: "Deep Double Descent via Smooth Interpolation"
_TMLR — Accepted by TMLR_

### Review · Reviewer_MGoA · 2022-12-29

**Summary Of Contributions:**

This paper proposes a way to calculate smoothness of neural networks, and  conducts empirical experiments to study the relationship among over-parameterization, smoothness, and testing performance.

**Audience:**

Yes

**Claims And Evidence:**

Yes

**Requested Changes:**

Please polish the paper to make it more clear, comparing with Belkin's work mentioned in the weakness part, and add some more experiments using other data sets.

**Strengths And Weaknesses:**

Strength:

[1] This paper clearly states the setups and observations of the experiments.

[2] This paper reveals a difference between clean data and noisy data (Section 4.5).

Weakness:

[1] The writing of this paper needs improvement.

  (1.1) The title "Deep Double Descent via Smooth Interpolation" from my understanding means that the paper developed a method of smooth interpolation and it caused the double descent phenomenon. This is different from the aim of this paper.

  (1.2) From the sentences in the abstract, "Common intuition..." and "While small interpolating ... in contrast to existing intuition", the authors study the smoothness in clean and corrupted samples and have some observations different from the existing literature. However, the authors put a lot of effort in introducing double descent phenomenon, e.g., "Our work presents an empirical study of deep double descent..." in Section 2. It is not clear why double descent is helpful to achieve the goal in the abstract.

I would suggest the authors to clarify what is the aim of this paper and how does double decent helps the understanding in this paper. I started to get confused when reading the introduction section.

[2] From the paper

Belkin, Mikhail, Daniel Hsu, and Ji Xu. "Two models of double descent for weak features." SIAM Journal on Mathematics of Data Science 2.4 (2020): 1167-1180.

my understanding is that, the model they considered have already implies that over-parameterized models some times smoothly interpolates all training data (including both noisy and clean). Although Belkin's work is on simple models rather than neural networks, it is not surprising that neural networks also share similar properties. I would suggest the authors to compare with this work.

[3] The numerical experiments use only CIFAR-10 and CIFAR-100. Since this paper only conducts empirical studies, using only these two datasets is not sufficient. In addition, the neural networks used in this paper are only ConvNet and ResNet, which is not sufficient.

[4] Some descriptions in the paper are not rigorous. For example, on Page 9, it is mentioned that "deep networks are biased towards learning training samples in approximately the same order". Please provide more concrete definitions of "bias" and "order".

[5] Some descriptions are not accurate. For example, in the abstract, it mentions that "Our findings show that loss sharpness in the input space follows both model- and epoch-wise double descent, with worse peaks observed around noisy labels". However, from Figure 7, noisy data does not have double descent.


Other issues:
[1] Is J(x,y) a vector or a matrix? If $L_\theta$ outputs a scalar loss, then how can we have the $||\cdot||_F$ norm of a vector? Please also double check if the words "Jacobian" and "Hessian" are used properly.

[2] Please adjust the layout of the paper. For example, Figure 2 appears on Page 6, but only until the end of Page 7 Figure 2 is mentioned.

[3] Please report the computation time for the proposed method in the experiments.

---

> ### Author Response · Authors · 2023-02-08
> **Inline response**
>
> We wish to thank you for your thorough review and for suggesting modifications to the paper. We reply to your comments inline below.
>
> ### Aim and message of the paper
> We begin by elucidating the connection between double descent and interpolation.
> The double descent phenomenon, as described by Belkin et al. (2019) is closely connected to the notion of interpolation. In fact, as the number of model parameters increases, models may overfit training data, with overparameterized models being capable of perfectly interpolating the training set. As model size increases even further, the resulting models will still interpolate training data, but their test error will start decreasing, thus causing a so-called second descent. Hence, the double descent phenomenon is intimately connected to interpolation.
> Next, we clarify the leading research question of our work. The aim of the paper is to better understand deep double descent, by studying it from the perspective of smooth interpolation of the training data, in relation to both model size and training epochs. Rather than proposing a novel interpolation algorithm, our focus is the study of how neural networks interpolate training data. Therefore, the two goals mentioned in your question, i.e., study of smooth interpolation or study of double descent, are both parts of our aim and in fact we show in this paper that they are closely related. Particularly, an open question is how networks that interpolate noisy data can generalise. Existing intuition built on polynomials and represented in Figure 1a, suggests that noisy labels are interpolated sharply, and that this may not affect generalisation as test data will lie outside such sharp peaks in which noisy labels are predicted.
> Our study proposes a quantitative metric for estimating sharpness of interpolation, and throughout our experiments we show that (1) sharpness of interpolation follows double descent, (2) large models smoothly predict noisy labels, (3) said smoothness is promoted by overparameterization, and in general it is not sufficient for generalisation.
> Therefore, our paper proposes a study of deep double descent through the lens of smoothness of interpolation, which could be an alternative title for our paper. In the current version of the manuscript, we opted for a shorter title that fits one line but it can be changed if it prevents central misunderstandings.
> We have updated the abstract and introduction of the paper to clarify the aim of the paper, as well as the relationship between double descent and interpolation.
>
> ### Missing related work
> Thank you for pointing out a related work that we had missed, which has now been discussed in the related works section. In summary, Belkin et al. (2020) propose models of double descent for the test error, assuming random Gaussian input and features, in the context of linear regression. Common to us the paper aims to study double descent for the test error. However, our works differ in that we study neural networks whose input dimension is fixed and independent from the sample size (unlike linear regression); we explicitly study smoothness of the loss landscape by the loss Jacobian and Hessian; we study interpolation beyond training data, by considering volumes around training samples; we separate fitting of cleanly-labelled samples from noisily-labelled ones; we also consider epoch-wise double descent. Finally, we note that all experiments in Belkin et al. involve the test error, while we study sharpness based on the training loss. Importantly, while the training loss does not follow double descent (see for instance Figures 2a and 4a), its Jacobian and Hessian do, which is on its own a novel and important finding.

---

> ### Author Response · Authors · 2023-02-08
> **Inline response. Part 2.**
>
> ### Additional models and datasets. Computational cost of our experiments.
> The experiments conducted in our work require significant computational resources. For each dataset and architecture, we train 64 different networks for 4000 epochs with three different seeds. We estimate the total computation time for the training of our models to have taken 6 GPU years on an NVIDIA Tesla A100. Furthermore, computing our statistics requires evaluating per-sample Jacobians for each training point and corresponding augmentations, for increasing volumes around each point. For each training setting, this was performed for 72 model checkpoints collected during training, to produce the heatmaps in Figures 5, 11, 12,13 and 14.
> In the revised version of the paper, we extend our experimental setup (appendix F.1)  to attention-based Transformers models (an additional architecture) trained on language modelling tasks (one additional task and two additional datasets), for which we estimate the Jacobian norm of the training cross entropy loss. Figure 10 extends our main finding beyond convolutional and vision models: input-space loss sharpness peaks near the interpolation threshold, and decreases afterward as model size grows.
> Computational resources are now detailed in appendix B, and appendix F.1 presents experiments beyond vision and convolutional models.
>
> ### Clearer definitions when referring to a related work.
> In section 4.2, we draw connections to related findings observed in prior work. Particularly the work of Hacoen et al. (2020) shows that neural networks share the order in which they learn samples throughout training. We have clarified the relation with Hacoen et al. (2020) in our revised version of the paper. However, due to the space limitation we can only refer the reader to that work, for a rigorous definition of “order”.
>
> ### Unclear sentence in the abstract, in relation to Figure 7.
> We agree that the sentence in the abstract is vaguely worded. Throughout our experiments, "peaks" of loss sharpness are observed consistently, with higher values on noisily-labelled samples. We see now that the abstract might appear as referring to the test error. The sentence has now been clarified.
>
> ### Clarification of notations.
> Since $\mathcal{L}_{\theta}$ is a scalar loss, $J(\mathbf{x},y) \in \mathbb{R}^{d}$ is a vector of the same dimensions as the input images, therefore it is appropriate to take a vector norm. We have adjusted our notation to clearly refer to the $\ell_2$ norm of vectors and matrices, with the latter being the standard Frobenius norm.
> Finally, we have checked all occurrences of Jacobian and Hessian and believe they are correct. However, if the reviewer has found inconsistencies that we didn’t notice, we would be happy to amend potential mistakes.
>
> ### Adjustments to layout of the paper.
> We have moved Figure 2 closer to its corresponding text.

---

> > ### Comment · Reviewer_MGoA · 2023-02-08
> > **Thanks for your response**
> >
> > I appreciate the authors' effort in revising the paper. Most of my concerns and comments have been addressed.
> >
> > About Belkin et. al. (2020):
> >
> > I understand that the proposed method in the submission is applied in neural networks while Belkin's work is for linear models. However, to intuitively explain why over-fitting in linear models does not hurt generalization, my understanding is that the interpolation do happens, but with only a vanishing probability measure. From this aspect, I was not surprised when reading the findings in this submission.
> >
> > In the authors' response, it is mentioned that the training loss always diminishes in the model size, while the double descent happens in the Hessian. If the loss landscape is flatter, the testing performance will be better. It is not surprising that the testing performance also gets a double descent curve.
> >
> > I agree with the novelty in this submission of proposing a metric and extending the analysis to neural networks.
> >
> > Since the proposed metric in this submission takes a long running time, it is acceptable for me not running other real-data experiments. I would suggest the authors to run some experiments in the high-dimensional linear regression setup so that the proposed method can cover both the scenarios in Belkin's work as well as neural networks. If the proposed method works in neural network, is it expected that it also works in linear regression models? If not, could you please provide some explanations?

---

> > > ### Author Response · Authors · 2023-02-10
> > > **Real data experiments and clarification on linear models**
> > >
> > > We thank the reviewer for their quick response. We are glad to see that most concerns are resolved by our initial response and revision. We reply inline below.
> > >
> > > ### Contextualization of our findings
> > > The work of Belkin et al. (2020) shows that, for the least squares solution to a noisy linear regression problem with random input and features, the impact of noise on generalization is mitigated by the abundance of weak features ($p > n$).
> > >
> > > We note that, for linear models, the output $y = \theta\mathbf{x}$ is a linear function of both parameters and input, and thus has zero curvature. Indeed, the only source of nonlinearity and curvature is the error function (MSE). Hence, as shown by Belkin et al., the error is ultimately controlled by the slope of the ground truth function, and the magnitude of noise, with the latter playing a different role according to the regime of parameterization. Particularly, the curvature of the error function is constant in the whole domain. Finally, we note that model size can only be controlled by changing the number of input features considered.
> > >
> > > In contrast, we study the case of nonlinear classification problems and nonlinear models, which have notable differences from the linear case. First, there is no a priori closed form solution of the learning problem, thus providing relevance to empirical studies. Second, curvature of the model function is non-constant, and the function may oscillate arbitrarily outside of the training data (this is known as the Runge phenomenon). Third, studies that rely exclusively on the test error suggest that interpolation is harmless also in overparameterized nonlinear models. Finally, the model function of convolutional architectures is independent of input-data dimensionality, and the relationship between complexity of the model function and its underlying parameterization is therefore implicit.
> > >
> > > In this setting, we experimentally show that, in the interpolating regime, (1) curvature at training points depends non-monotonically on model size; (2) oscillations occur especially for small interpolating models, which are worst affected by noise; (3) large models achieve low-curvature interpolation of both clean and noisy samples (in contrast with the polynomial intuition), and such property is observed over large volumes (non-zero measure) around each training point (in contrast with the Runge phenomenon, thus providing evidence of implicit regularization); (4) Interpolation of noise impacts generalization even for large models (contrary to the overparameterized linear regression case); (5) Double descent observed for input space curvature occurs even when fitting 100% noisy data, more clearly pinpointing properties that are consistently promoted by overparameterization in deep nonlinear networks.
> > >
> > > ### Clarification on the suggested experiments for linear models
> > > Our methodology enables the study of sharpness of fit of training data for nonlinear models, providing a comparative study of the regularity with which different parameterizations achieve interpolation and (in some cases) generalization. For linear models, our quantities can be computed in closed form without resorting to numerical simulations, and the mean-squared-error presents constant curvature related to the slope of the linear model considered. Hence, we are unsure about what quality of linear models the suggested experiment should be designed to capture, and we would appreciate the reviewer’s input in designing an experiment that could complement our study.
> > >
> > > ### Real data experiments
> > > We thank the reviewer for acknowledging the computational complexity of our experiments. In our initial response, we have included additional experiments on real-world datasets and architectures (Transformers trained on two machine translation tasks). Our results can be found in section F.1.

---

> > > > ### Comment · Reviewer_MGoA · 2023-02-12
> > > > **Linear models**
> > > >
> > > > Thanks for your response.
> > > >
> > > > It is now more clear for me the difference between this submission and Belkin's previous work in the linear models. I appreciate the authors explaining these details. I would suggest the authors put the above illustrations in the paper, e.g., in the appendix. I think this information is important to researchers working in theories.
> > > >
> > > > For the experiment for the linear models, the authors mention that "our quantities can be computed in closed form without resorting to numerical simulations, and the mean-squared-error presents constant curvature related to the slope of the linear model considered". I would suggest the authors also refine the wordings and put it in the paper.

---

> > > > > ### Author Response · Authors · 2023-02-15
> > > > > **Changes implemented**
> > > > >
> > > > > We would like to thank the reviewer for their feedback. The proposed changes have now been implemented in section F of the supplemental material (appendix).

---

> > > > > > ### Author Response · Authors · 2023-03-04
> > > > > > **Updating rating of the submission**
> > > > > >
> > > > > > We hope that the changes implemented in the latest revision of our paper of Feb 15 &mdash; namely discussing the relationship and differences between our study and linear models &mdash; address the points raised in your review. If you find our revision satisfactory, could you please update the "Claims and Evidence" rating in your original review?

---

### Review · Reviewer_5jmG · 2023-01-12

**Summary Of Contributions:**

This paper performs an empirical study of the smoothness of the loss landscape of DNNs in the input space. The key findings include:

(1) overparameterization improves the input-space smoothness
(2) the double descent phenomenon can be observed with respect to both training epochs and model sizes.

Overall, this paper highlights the importance of studying the smoothness of the input-space to gain a better understanding of the implicit regularization effect of overparameterization.

**Audience:**

Yes

**Broader Impact Concerns:**

No broader impact concerns.

**Claims And Evidence:**

Yes

**Requested Changes:**

Please refer to the weakness section.

**Strengths And Weaknesses:**

Strength:

1. The novelty is good, studying the effect of overparameterization and its relationship to the generalization via characterizing the input-space smoothness is a nice direction.
2. The finding is also interesting: the overparameterization helps reduce the sharpness of the input space.
3. The idea of characterizing the smoothness via estimating the geodesic paths is also interesting and potentially useful.

Weakness:

1. Lack of comparison. It seems that the authors try to argue that the input-space smoothness could be better than the parameter-space smoothness. To better explain this, it would be better to draw similar plots as Figures 2&3 in terms of Jacobian/Tangent Hessian metrics for model parameters. Then we are able to see whether overparameterization can also encourage the parameter-space smoothness.

2. The experiment setup is not clearly presented. For instance, the authors may need to explain what's the number of augmentations in Figure 2. I can understand that this will affect the estimations of jacobian and Tangent Hessian, but why the crossentropy will also be different, do you use augmented data to calculate the loss? Or do you use the augmented data to train the model?

3. In order to show the double descent, it would be good to also include the test error/loss.

4. It seems that the sharpness will be largely different if using different numbers of augmentations. However, this is confusing to me as it seems that using different numbers of augmentations will also affect the estimation of sharpness, which implies they should be at least in the same order since the true sharpness of the loss does not depend on the estimating approaches.

5. Regarding Figure 7, the authors claim that the double descent for the corrupted cannot be observed. However, I actually do not see the difference between the trends for the clean and corrupted classes. Could you clarify this with more details?

---

> ### Author Response · Authors · 2023-02-08
> **Inline response**
>
> We would like to thank you for your thorough feedback, and for pointing out ambiguities and suggesting clarifications. Please find below point-by-point replies.
>
> ### Comparison with parameter-space analysis of the loss landscape.
> In the following, we briefly summarise the advantages of input-space vs parameter-space studies for addressing our research question.
> Our goal is to study the emergence of double descent – which occurs for interpolating models – through the lens of smoothness of fit of training data, with the aim of exploring complexity of neural network functions trained in practice, and how they interpolate data (see Figures 1a and 1b for a toy example of sharp vs smooth interpolation respectively).
> We interpret neural networks as functions $\mathbf{f}: \mathcal{X} \times \Theta \to \mathcal{Y}$, with parameter $\theta$ and input $\mathbf{x} \in \mathcal{X}$.
> A large body of work focuses on representations, thus studying $\theta$ in relationship to optimization, and parameter-space curvature of the loss landscape. These works aim to study properties of minima $\theta^*$, as well as when and how such solutions are recovered. While this line of work is of great practical value, and produced state-of-the-art optimizers, such studies are confounded by weight-space symmetries (e.g. permutation invariance of linear layers), for which the same neural network function might be expressed by different values of $\theta$. Especially, studies of parameter-space curvature do not allow for a direct analysis of interpolation at the level of individual training points.
> Therefore, in our work, we take a different view of neural networks, interpreted as functions $\mathbf{f}_{\theta}: \mathcal{X} \to \mathcal{Y}$, which decouples the study from specific parametrizations of the hypothesis.
> Importantly, this allows us to study directly interpolation of training data, including separating cleanly- from noisily-labelled points, as well as the behaviour of the network function in neighbourhoods of the training data.
> We have reworded the introduction and related works of our submission to better frame our methodology w.r.t. parameter-space studies of the loss landscape.
>
> ### Use of data augmentation for Monte Carlo integration.
> The number of augmentations $K$ controls the volume of integration, while the number of paths $P$ is the number of MC samples used for the volume integration. Therefore all metrics considered, namely Jacobian, tangent Hessian, accuracy, and cross entropy can change as we increase the number of augmentations and thereby the volume. In the following, we clarify the use of data augmentation in our methodology, as well as in our experiments.
> First, we wish to note that all networks in our experimental study, except for Figure 6, are trained without any form of data augmentation.
> Next, we summarise the aim of our proposed geodesic integration (Section 3). For each training point, we wish to estimate how each considered metric (cross entropy, accuracy, Jacobian and tangent Hessian norms) changes over volumes around each training point, as volume size increases.
> Specifically, for each training point, we construct $P$ geodesic paths emanating from that point, and estimate each metric over volumes by Monte Carlo integration over the $P$ paths (Figure 1c and 1d). Intuitively, the length of each path – specified by the number of augmentations $K$ used to generate the path – corresponds to the radius of the volume considered, and the number $P$ of paths corresponds to the number of Monte Carlo samples used to estimate the volume. Throughout our experiments, we keep $P$ fixed, i.e. the number of Monte Carlo samples is the same for each and every experiment.
> Finally, we take a mean-field view, by averaging the volume-based measure over the training set. This corresponds to Equation 7.
> When increasing the volume of integration of a given metric, e.g. cross entropy, one might expect the loss over larger volumes to increase as one moves away from a training point, as neural networks are trained to minimise the loss only at the training points. Indeed, near the interpolation threshold, networks fit training points "sharply", and the loss increases as one moves away from the training data (Figure 1c). For large models,  we see that the loss, as well as Jacobian and Hessian, remain low as one moves away from each training point, showing that data is interpolated smoothly (Figure 1d). Hence, "sharpness of interpolation" follows double descent, and we observe that said trend strongly correlates with double descent for the test error.
> We clarified our methodology by: (1) providing an illustration of geodesic integration in Figure 1; (2) Providing pseudocode for generating geodesic paths in appendix D; (3) Clarifying section 3.3; (4) Updating the legends of Figures 2-4.

---

> ### Author Response · Authors · 2023-02-08
> **Inline response. Second part.**
>
> ### Clearer reference to test error/loss.
> In the original version of the manuscript, test error and loss for each network/dataset considered were reported in Figures 1c, 1d, 6a, and 7b. The test error and loss curves have now been merged with Figures 2-4.
>
> ### Dependency of volume-based sharpness on the number of augmentations.
> Upon revising the paper, we believe the notation used in the legends of Figures 2-4 and 6-7 was prone to confusion. The number $K$ of augmentations is supposed to denote the radius of the volume considered, based on the length of the geodesic paths, which is represented by the number of augmentations used to generate it. The number of MC samples set, i.e., the number $P$ of geodesic paths, is constant for all of the plots. We believe the modifications in the current revision based on question 2 should clarify this issue as well.
> Throughout our experiments, it can be observed that sharpness peaks near the interpolation threshold, while for larger models sharpness decreases, as indicated by each volume-integrated measure (Equation 6) approaching zero as model size increases.
> We have updated the legends of Figures 2-4 to clarify that each line plot refers to different volumes of integration, rather than different number of MC samples.
>
> ### Clarifications on Figure 7
> All experiments until Figure 7 and section 4.5 are focused on establishing a strong correlation between smooth interpolation and double descent for the test error. In Figure 7, we propose a simple experiment to decouple double descent in loss landscape sharpness from double descent of the test error.
> Specifically, by corrupting labels only for selected classes, we aim to explore whether either smoothness, generalisation (or both), emerge from overparameterization. To do so, we devise an experimental setting in which smooth interpolation of noisy data does not result in improved generalisation. Particularly, we corrupt labels only of selected classes, train all networks to interpolate the training set, and then study loss sharpness for clean and corrupted training samples. However, this time we also split the test set into two subsets, namely test samples belonging to classes which were perturbed at training time, and the rest. We note that only training samples were noisily labelled during training, and the test points and the respective ground truth labels are used for evaluating the test error, as normal practice.
> In this experimental setting, we observe that loss landscape sharpness still follows double descent, for both cleanly- and noisily-labelled training samples, thus showing that overparameterization promotes smooth interpolation of the training data (Figure 7a). At the same time, smooth interpolation results in improved generalisation for large models only for the classes that were not perturbed at training time (Figure 7b, top). In fact, no double descent for the test error is observed for the test samples belonging to classes that were heavily corrupted at training time.
> We have carefully reworded section 4.5, to clearly describe trends observed in practice.

---

### Review · Reviewer_egGL · 2023-01-30

**Summary Of Contributions:**

The paper provides empirical study of the landscape smoothness in the overparameterized regime of neural networks. Through computing the Jacobian and tangent hessian at training data points, the paper finds that after reaching the interpolation threshold, neural networks tend to smoothly interpolate data, rather than forming spikes at training data points. This discovery enhances the understanding of neural networks' behavior in the practical setting (overparameterized). The reviewer believes this is a timely study and interesting discovery regarding overparameterized neural networks.

**Audience:**

Yes

**Broader Impact Concerns:**

No ethical concerns.

**Claims And Evidence:**

Yes

**Requested Changes:**

I would suggest the authors to read through the manuscript once more to clear a few typos and grammatic issues.

**Strengths And Weaknesses:**

The paper is clearly motivated and relatively easy to follow. Empirical results are rich and are performed under relevant and interesting settings. The following questions might be relevant.

1). Is 20% label corruption needed to study the smoothness of interpolation? Or changing different levels of label corruption, can we find different conclusions? As a sanity check, if the labels are totally corrupted, does overparameterized networks still produce smooth interpolations?

2). How accurate is the approximation to the tangent hessian estimation? As we know, data are high dimensional and in order to approximate a high-dimensional integral, exponentially many (exponent depending on data dimension, also known as the curse of dimensionality) samples are needed.

3). The finding of smooth interpolation contradicts previous intuition. Can authors comment on the cause of smooth interpolation? Is it a consequence of algorithmic regularization in SGD or is it an unique feature of large neural networks? The paper is expository yet a bit weak in explaining the thinking behind the smooth interpolation phenomenon. The paper should be stronger with more discussions.

---

> ### Author Response · Authors · 2023-02-08
> **Inline response**
>
> We would like to thank you for your thorough feedback, and for suggesting additional experiments. Please find below point-by-point replies.
>
> ### Impact of label noise on CIFAR-10.
> The amount of label noise used for the CIFAR-10 experiments is in general not needed when double descent for the test error is pronounced.
> When designing our experimental setup, we decided to focus on settings that show a clear double descent in the test error, in order to study the problem through the lens of smoothness of interpolation. For CIFAR-10, corrupting a small fraction of the training labels was needed in order to produce a marked double descent curve for the test error. Furthermore, artificially perturbing labels allows us to study interpolation of cleanly- and noisily- labelled points separately.
> For CIFAR-100 (Figures 1d and 4), a marked double descent for the test error can be observed even without corrupting any fraction of the training labels. We hypothesise this arises from CIFAR-100 being a more complex dataset, with only 500 samples per class, as opposed to 5000 samples per class in CIFAR-10.
>
> ### Smooth interpolation of randomly labelled data.
> This is indeed a very interesting question to study, which in the spirit of Figure 7a, could further confirm the emergence of "smooth interpolation" decoupled from double descent in the test error. We expect that, if a model is large enough to interpolate the training set, overparameterization should promote smooth interpolation. One can see evidence of this phenomenon in Figure 7a and 7b (top). When training labels are heavily corrupted, double descent in loss sharpness follows double descent, while the phenomenon no longer occurs for the test error for the corrupted classes. Hence, smoothness – promoted by overparameterization – is a model prior that is not always aligned with improved generalisation.
> To further test this hypothesis, following your suggestion, we trained a family of ResNet18s of increasing width on CIFAR-10 with 100% training labels corrupted, and estimated sharpness of interpolation via the Jacobian norm, for increasing volumes around each training point.  The observations are reported in Figure 7b (bottom), which confirms that overparameterization promotes smooth interpolation, and that this may not be aligned with generalization.
>
> ### Tangent Hessian approximation and curse of dimensionality.
> Our choice of estimating the input-space Hessian by its tangent approximation is indeed motivated by the fact that Monte Carlo sampling suffers from the curse of dimensionality. Our work proposes Hessian tangent estimation as a specialisation of Hessian eigenmaps (Donoho & Grimes, 2003) and Rugosity (LeJeune et al., 2019) to the case of image data. Particularly, the method scales linearly in the number $d$ of dimensions of the Hessian. While for the CIFAR datasets $d$ is the order of 9M, considering the tangent Hessian reduces the dimensionality of the problem to that of the data "manifold" local to each point. For CIFAR, the dimensionality of the input data was empirically estimated by Pope et al. (2021)  to be in the order of $d_{\mathcal{M}} = 30$. By using weak colour augmentations (which lie on-manifold), as well as the whole training set, the mean Hessian is estimated by using about $250$k points, which is well above the estimated intrinsic dimensionality of the datasets considered. Importantly, the measure considered is able to track double descent, and so is sufficient for comparing relative trends.
> Reference: Pope, P., et al. "The Intrinsic Dimension of Image Data and Its Impact on Learning". ICLR’21.
>
> ### Strengthened conclusions, discussing potential causes of smooth interpolation.
> We believe this is indeed a very important question, that is currently under active research. On the one hand, several studies point out that "stochastic" optimization, and the potential implicit regularisation effect of mini-batch noise, are not required for generalisation. On the other hand, there is no conclusive answer for neural networks at the moment, as current models of double descent hypothesise that stochastic noise is an important component in explaining implicit regularisation in deep learning.
>
> Finally, as double descent is a general phenomenon affecting overparameterized models (see Belkin et al. (2019) for an overview), we believe the phenomenon is not tied to neural networks exclusively. Indeed, our study treats neural networks as general functions $\mathbf{f}: \mathcal{X} \to \mathcal{Y}$ without considering the role of layers hierarchy, so we expect a general theory to capture neural networks as well as other classes of models.
> We have expanded the conclusions section of our manuscript, to discuss smoothness in relationship to model architecture and optimization.

---

### Author Response · Authors · 2023-02-08
**General response**

We would like to thank all reviewers for their thorough reviews and constructive comments.
We are happy you found our research direction to study double descent through the lens of input-space smoothness to be novel (5jmG), timely (egGL), interesting (5jmG, egGL), presenting clear observations (M5GoA) as well as a geodesic integration method with potential relevance beyond our study (5jmG).
Below, we summarise the main changes in the updated revision of the manuscript, and we address each reviewers’ concerns by replying to the corresponding threads.

### Changes in Revised Version
Here, we summarise the main changes in the revised manuscript.
- Abstract: clearer connection between interpolation and double descent (MGoA).
- Introduction: Clarified our research question (MGoA); Outlined advantages of input-space studies of the loss landscape for addressing our research question (5jmG).
- Related work: Clearer contextualisation of our work in connection to empirical studies of the loss landscape of neural networks in parameter space as well as input space (5jmG); Added reference to missing related work of Belkin et al. (2020), (MGoA).
- Methodology: clearer description of the use of data augmentation in geodesic volume-based integration (5jmG), with reference to pseudocode for generating geodesic paths, as well as an illustration of volume based integration via geodesic paths; Clarified mathematical notation with respect to vector norms (MGoA).
- Experiments: Added experiment with networks trained to fit 100% noisy training labels (egGL); updated legends of Figures 2-4 to refer to the volumes considered during Monte Carlo integration (5jmG); Updated caption of Figures 2-4 to clearly refer to corresponding double descent curves for the test error (5jmG); Clarified reference to related works in section 4.2 (MGoA); Clarified analysis of the observed trends in section 4.5 and Figure 7, for which smooth interpolation of noisy data does not always result in double descent of the test error (5jmG).
- Conclusions: Added discussion of role of model size and optimizer in promoting smooth interpolation of training data for overparameterized models (egGL);
- Appendix: Added description of computational resources required to reproduce our experiments (MGoA); Included additional architecture, learning task, and datasets (MGoA).

---

### Decision · Action_Editors · 2023-03-13

**Recommendation:** Accept as is

**Comment:**

In this paper, a method for computing the smoothness of neural networks is proposed and empirical experiments are conducted to investigate the relationship between overparameterization, smoothness, and test performance. The paper finds that the investigated neural networks tend to interpolate data smoothly after reaching the interpolation threshold instead of forming spikes at the training data points. This discovery improves the understanding of overparametrized neural networks and can be seen as a way to extend the existing theoretical analysis on the double descent phenomenon to neural networks.

The reviewers cautiously note that the numerical study may not be comprehensive enough to make generalizable statements for other networks/datasets. In addition, the metrics measurement methods chosen may have an impact on the transferability of the results to other applications. Despite these reservations, the reviewers believe that this is a timely study that meets the acceptance criteria of the TMLR.

**Audience:**

Yes. The phenomenon of double descent has so far been theoretically analyzed mainly on convex problems. The empirical investigation carried out in this paper may be an important step towards transferring the theoretical findings to overparameterized neural networks.

**Claims And Evidence:**

Yes. The paper carefully describes the experimental setup and how the relevant metrics are calculated and reported. The claims are verified on two image datasets (CIFAR-10 and CIFAR-100) with two networks (ConvNet and ResNet).